# Estimating Noise Correlations Across Continuous Conditions With Wishart Processes

**Amin Nejatbakhsh**      **Isabel Garon**      **Alex H Williams**
Center for Neural Science, New York University, New York, NY
Center for Computational Neuroscience, Flatiron Institute, New York, NY
{anejatbakhsh,igaron,awilliams}@flatironinstitute.org

## Abstract

The signaling capacity of a neural population depends on the scale and orientation of its covariance across trials. Estimating this "noise" covariance is challenging and is thought to require a large number of stereotyped trials. New approaches are therefore needed to interrogate the structure of neural noise across rich, naturalistic behaviors and sensory experiences, with few trials per condition. Here, we exploit the fact that conditions are smoothly parameterized in many experiments and leverage Wishart process models to pool statistical power from trials in neighboring conditions. We demonstrate that these models perform favorably on experimental data from the mouse visual cortex and monkey motor cortex relative to standard covariance estimators. Moreover, they produce smooth estimates of covariance as a function of stimulus parameters, enabling estimates of noise correlations in entirely unseen conditions as well as continuous estimates of Fisher information—a commonly used measure of signal fidelity. Together, our results suggest that Wishart processes are broadly applicable tools for quantification and uncertainty estimation of noise correlations in trial-limited regimes, paving the way toward understanding the role of noise in complex neural computations and behavior.

## 1   Introduction

Nominally identical repetitions of a sensory stimulus or a behavioral action often elicit variable responses in neurons. Intuitively, this "noise" degrades the information capacity of neural representations. However, the precise nature of this effect is complex since neural responses are not independent random events. Indeed, pairs of co-recorded neurons exhibit significant "noise correlations" from trial-to-trial, with well-studied and intriguing consequences [1–5]. In particular, introducing noise correlations into a circuit can theoretically degrade or enhance its signaling capacity depending on the geometric configuration of neural representations in firing rate space (Fig. 1A; [6, 7]).

Estimating noise covariance is therefore an important problem that arises frequently in neural data analysis.[1] However, this problem is widely regarded as challenging [8–10] because the number of estimated parameters (i.e., number of unique neuron pairs) grows quadratically with the number of co-recorded neurons, while the number of measurements per trial (i.e., number of neurons with a response) grows only linearly (for more discussion see, e.g., [11]). As a result, experimental investigations into noise correlation have been limited to extremely simple tasks where a very large number of trials can be collected over a small number of conditions. For example, Rumyantsev et al. [10] recorded neural responses over just two stimulus conditions (oriented visual gratings), repeated across hundreds of trials. We therefore have a rudimentary characterization of how noise correlations

---

[1]Throughout, we focus on the problem of estimating noise *covariance* instead of noise *correlation*. By accurately estimating the former, we can also estimate the latter since the correlation between two neurons is given by their covariance normalized by each neuron's marginal variance.

37th Conference on Neural Information Processing Systems (NeurIPS 2023).

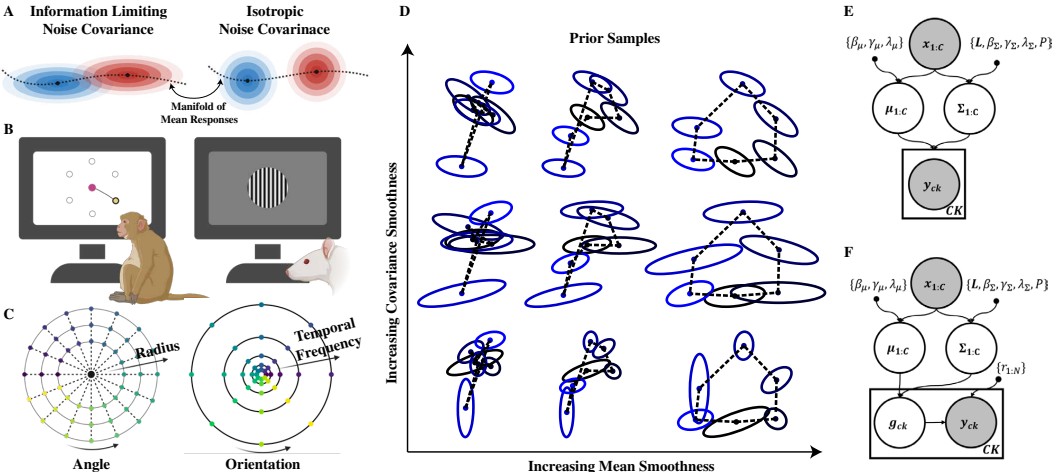

**Figure 1:** (A) Illustration of information limiting noise correlations. (B) Experimental datasets with smoothly parametrized conditions (see [7]). *Left*, a nonhuman primate makes point-to-point reaches to radial targets. *Right*, a rodent views drifting visual grating stimuli of different orientations and temporal frequencies (i.e. speed). (C) Parameterized condition spaces for the datasets in B. (D) Samples from Gaussian and Wishart process prior distributions for with $C = 6$ conditions over 1D periodic variable (e.g. angle) and $N = 2$ neurons at varying kernel smoothness parameters ($\lambda$ in eq. 6). Dots and ellipses represent the mean and covariance of neural responses. Colors appear in the increasing order of condition value (e.g. angle) and dashed lines connect neighboring conditions. Increasing $\lambda_\mu$ (horizontal axis) encourages smoothness in the means while increasing $\lambda_\Sigma$ (vertical axis) encourages the ellipses to change smoothly in scale and orientation. (E-F) Graphical model of the Wishart model with Normal (E) and Poisson (F) observations (see Supplement B.5 for details of the Poisson model).

change across rich or naturalistic experimental conditions, despite interest among the theoretical community [5, 12, 13], and experimental evidence that noise statistics are context-dependent [14–16]. New statistical approaches that are capable of inferring noise covariance with fewer trials per condition are needed to make progress on this subject.

To address this need, we develop a probabilistic model based on Wishart processes [17]. The core insight we exploit is that similar experimental conditions—e.g. visual gratings at similar orientations or cued arm reaches to similar locations—ought to exhibit similar noise statistics. Thus, by building this insight into the prior distribution of a probabilistic model, we can pool statistical power across neighboring conditions and accurately infer noise correlation structure with fewer trials per condition. In fact, this principle also enables us to predict covariance in entirely unseen conditions by interpolating between (and potentially extrapolating from) observed conditions. We demonstrate the scientific potential and generality of this approach through two experimental applications spanning different modalities (vision and motor tasks) and species (mouse and nonhuman primate).

## 2 Methods

### 2.1 Problem Setup

Consider an experiment with $N$ neurons recorded over $C$ distinct stimulus conditions, each with $K$ repeated trials.[2] Experimental advances continually enable us to record larger neural populations (large $N$) and neuroscientists are increasingly interested in performing rich and naturalistic experiments (large $C$). Both of these trends put us in a regime where $K < N$, meaning there are few trials per condition relative to the number of neurons [9].

---

[2]To simplify exposition, we assume that there are an equal number of trials in each condition. However, the model easily applies to the more general case with unbalanced trial allocations.

**Empirical Estimator:** Our goal is to estimate the $N \times N$ covariance matrix, $\mathbf{\Sigma}_c$, describing trial-by-trial variability in each condition $c \in \{1, \ldots, C\}$. If $\mathbf{y}_{ck} \in \mathbb{R}^N$ is the measured response on trial $k$ of condition $c$, then a straightforward estimator is the empirical covariance:

$$\overline{\mathbf{\Sigma}}_c = (1/K) \sum_k (\mathbf{y}_{ck} - \overline{\mathbf{y}}_c)(\mathbf{y}_{ck} - \overline{\mathbf{y}}_c)^\top , \quad \text{where} \quad \overline{\mathbf{y}}_c = (1/K) \sum_k \mathbf{y}_{ck} . \tag{1}$$

This estimator behaves poorly in our regime of interest when $K < N$. Indeed, while the true covariance must be full rank, the empirical covariance is singular with $N - K$ zero eigenvalues.

**Regularized Estimators:** Statisticians often refer to problems where $K < N$ as the *high-dimensional regime*. Estimating covariance in this regime is actively studied and understood to be challenging [18]. The Ledoit-Wolf estimator [19] is a popular approach, which regularizes ("shrinks") the empirical covariance towards a diagonal estimate. Other estimators shrink the individual covariance estimates towards the grand covariance (see eq. 2 below) instead of a diagonal target [20]. Graphical LASSO [21] is another popular approach, which incorporates sparsity-promoting penalty on the inverse covariance matrix. All of these methods can be understood as forms of regularization, which reduce the variance of the covariance estimate at the expense of introducing bias.

**Grand Empirical Estimator:** The methods summarized above (or variants thereof) are common within the neuroscience literature [8, 22, 23]. It is also common to simply estimate the covariance as being constant across conditions [10, 24], in which case an estimate can be found by pooling trials across all conditions to achieve an empirical estimate of the grand covariance:

$$\overline{\overline{\mathbf{\Sigma}}} = \frac{1}{C} \sum_c \overline{\mathbf{\Sigma}}_c = \frac{1}{KC} \sum_{k,c} (\mathbf{y}_{ck} - \overline{\mathbf{y}}_c)(\mathbf{y}_{ck} - \overline{\mathbf{y}}_c)^\top . \tag{2}$$

This estimator can be thought of as having high bias (since covariance is often thought to be stimulus-dependent [14–16]), but low variance since many more trials can be averaged over after pooling across conditions. We compare our model to these baselines in section 3.

## 2.2 Wishart Process Model of Noise Correlations

Our insight is to incorporate an alternative form of regularization whenever the experimental conditions are smoothly parameterized. For instance, a classical stimulus set used by visual neuroscientists is a grating pattern varied over $C$ orientations. In fact, such stimuli are commonly used for noise correlation estimation [8, 10, 25, 26]. Intuitively, two gratings with nearly identical orientations should evoke similar probabilistic responses. Thus, as stimulus orientation is incremented, the mean and covariance of the response should smoothly change along a manifold (Fig. 1A).

This idea readily extends to other experimental settings. For example, pure acoustic tones with smoothly varied frequencies are often used in auditory neuroscience [27]. In studies of olfactory coding, isolated odorants or mixtures of odorants can be presented at smoothly varied concentrations [28]. In motor neuroscience, voluntary reach movements can be cued to continuously varied target locations [29]. Finally, in the context of navigation, noise correlations have also been studied in neural ensembles coding for continuous variables like position [30] and head direction [31]. In principle, the modeling approach we develop is applicable to any of these experimental settings. This is in contrast to experimental settings that employ a discrete set of stimuli with no underlying continuous parameterization (such as olfactory responses to distinct classes of odors).

To formalize these concepts, let $c \in \{1, \ldots, C\}$ index the set of conditions, and let $\mathbf{x}_c \in \mathcal{X}$ denote the stimulus or animal behavior in condition $c$. For example, $\mathbf{x}_c$ could be a scalar denoting the orientation of a voluntary reach or a visual grating (Fig. 1B). However, the model we will describe is more general than this: $\mathbf{x}_c$ could be a vector describing a multi-dimensional stimulus space. For example, we will later study motor preparatory activity to reach targets at different radii and angles (sec. 3.2). In this setting, $\mathbf{x}_c$ is a 2-dimensional vector specifying these two parameters of the reach target (Fig. 1C, left). We will also study neural responses to drifting visual gratings (sec. 3.3). In this setting, $\mathbf{x}_c$ is a vector specifying the speed and orientation of the grating stimulus (Fig. 1C, right).

Recall that $\mathbf{y}_{ck} \in \mathbb{R}^N$ denotes the simultaneously measured response of $N$ neurons to condition $c \in \{1, \ldots, C\}$ on trial $k \in \{1, \ldots, K\}$. Our goal is to model these measured responses as samples

from some distribution $F$ with mean $\boldsymbol{\mu}$ and covariance $\boldsymbol{\Sigma}$:

$$\boldsymbol{y}_{ck} \sim F(\boldsymbol{\mu}(\boldsymbol{x}_c), \boldsymbol{\Sigma}(\boldsymbol{x}_c)) \quad \text{independently for } c, k \in \{1, \ldots C\} \times \{1, \ldots K\}. \tag{3}$$

Importantly, the functions $\boldsymbol{\mu} : \mathcal{X} \mapsto \mathbb{R}^N$ and $\boldsymbol{\Sigma} : \mathcal{X} \mapsto \mathbb{R}^{N \times N}_{++}$ should be smooth so that the response distribution changes gradually as the experimental condition is altered. To accomplish this, we leverage the machinery of Gaussian processes, which can be used to specify a prior distribution over smooth functions [32]. While Gaussian process models have been extensively used within statistical neuroscience [33–37], they have typically been used to model smoothness in temporal dynamics rather than smoothness across experimental conditions (but see [38]).

In this paper, we assume that $F(\boldsymbol{\mu}, \boldsymbol{\Sigma})$ is a multivariate Gaussian distribution, which enables tractable and convenient probabilistic inference (see sec. 2.4). Since many noise correlation analyses are applied to spike count data, it is of great interest to extend our model so that $F$ is better suited to this case. Multivariate extensions of the Poisson distribution are an attractive option [39]. While the focus of this paper is on multivariate Gaussian distributions, we develop inference tools for the Poisson model and we leave further investigation of this model for future work (see 1F for the corresponding graphical model and Supplement B.5 for further details). We note that it is common to pre-process spike count data and calcium fluorescence traces (e.g. through square root or log transforms) so that they are better approximated by Gaussian models [33, 40, 41].

To model $\boldsymbol{\mu}(\cdot)$, we stack independent Gaussian processes into an $N$-vector. To model $\boldsymbol{\Sigma}(\cdot)$, we use a *Wishart process* [42]. In essence, the Wishart process (denoted by $\mathcal{WP}$) stacks Gaussian processes into an $N \times P$ matrix and uses this as an overcomplete basis for the covariance. Formally,

$$\boldsymbol{\mu}(\cdot) \sim \mathcal{GP}^N(\boldsymbol{0}, k_\mu), \quad \boldsymbol{U}(\cdot) \sim \mathcal{GP}^{N \times P}(\boldsymbol{0}, k_\Sigma), \quad \boldsymbol{\Sigma}(\boldsymbol{x}) = \boldsymbol{U}(\boldsymbol{x})\boldsymbol{U}(\boldsymbol{x})^\top, \tag{4}$$

where $P \geq N$ is a hyperparameter, and $\mathcal{GP}^N(\boldsymbol{m}, k)$ denotes an $N$-dimensional Gaussian process with (constant) mean $\boldsymbol{m} \in \mathbb{R}^N$ and a positive definite kernel function $k : \mathcal{X} \times \mathcal{X} \mapsto \mathbb{R}$, described below. For any fixed value of $\boldsymbol{x} \in \mathcal{X}$, the product $\boldsymbol{U}(\boldsymbol{x})\boldsymbol{U}(\boldsymbol{x})^\top$ follows a Wishart distribution, which is a standard probabilistic model for covariance and inverse covariance (see sec. 3.6 of [43]). To our knowledge, Wishart processes have not previously been leveraged for neural data analysis.

The kernel functions $k_\mu$ and $k_\Sigma$ play important roles by respectively specifying the degree of smoothness in the mean and covariance of the neural response across conditions (Fig. 1D). The kernel functions must satisfy certain technical conditions, as reviewed in [44, 45]. We satisfy these conditions by defining a kernel for each coordinate of $\boldsymbol{x} \in \mathcal{X}$ and evaluating the product, $k(\boldsymbol{x}, \boldsymbol{x}') = \prod_i k_i(x_i, x_i')$. This is guaranteed to produce a valid kernel ([44], Proposition 13.2). For non-periodic variables, we use a standard squared exponential kernel

$$k(x, x') = \gamma \delta_{xx'} + \beta \exp\left[-(x - x')^2/\lambda\right] \tag{5}$$

where $\{\gamma, \beta, \lambda\}$ are hyperparameters and $\delta_{xx'} = 1$ when $x = x'$ and otherwise evaluates to zero. For periodic conditions (e.g. orientation of visual gratings or radial reach directions) we use [46]:

$$k(x, x') = \gamma \delta_{xx'} + \beta \exp\left[-\sin^2(\pi|x - x'|/T)/\lambda\right] \tag{6}$$

where $T$ specifies the period of the kernel. For both kernels, the most critical hyperparameter is $\lambda$ (often called the "bandwidth"), which specifies the degree of smoothness across conditions. The remaining hyperparameters, $\gamma$ and $\beta$, respectively ensure that the Gaussian process is well-conditioned and scaled. We tune all of these hyperparameters to maximize heldout log-likelihood by cross-validation. Throughout, we use $\{\gamma_\mu, \beta_\mu, \lambda_\mu\}$ and $\{\gamma_\Sigma, \beta_\Sigma, \lambda_\Sigma\}$ to respectively represent the kernel hyperparameters for $k_\mu$ and $k_\Sigma$.

## 2.3 Low-Rank Plus Diagonal Covariance and Other Model Variants

The vanilla Wishart process model in equations (3) and (4) can be extended in several ways, which we found to be useful in practical applications. First, we can augment the covariance model by adding a diagonal term, resulting in a model that is reminiscent of factor analysis:

$$\boldsymbol{\Sigma}(\boldsymbol{x}) = \boldsymbol{U}(\boldsymbol{x})\boldsymbol{U}(\boldsymbol{x})^\top + \boldsymbol{\Lambda}(\boldsymbol{x}), \tag{7}$$

where $\boldsymbol{\Lambda}(\boldsymbol{x}) \in \mathbb{R}^{N \times N}$ is a condition-dependent nonnegative diagonal matrix where the diagonal terms are independent samples from a $\mathcal{GP}^N$ that are transformed using a `softplus` function,

$f(x) = \log(1 + e^x)$, to ensure positivity. We term this extension of $\mathcal{WP}$ as $\mathcal{WP}^{lrd}$ (for low-rank plus diagonal). Intuitively, this model extension has fewer parameters than a vanilla $\mathcal{WP}$ and hence discourages overfitting. Additionally, the diagonal term, $\mathbf{\Lambda}(\boldsymbol{x})$, helps ensure that the covariance matrices remain well-conditioned during inference. Indeed, we can choose values of $P < N$ (or even $P = 0$), which enables us to specify prior distributions over the covariance that are "low-dimensional" in the sense that a large fraction of variance can be explained by a small number of dimensions. This comports with numerous experimental observations [10, 47]. As can be seen in Supplement Fig. 5, the dimensionality of matrices sampled from a vanilla Wishart distribution scales linearly with $N$, while matrices sampled according to equation (7) saturate at lower values.

Next, we additionally incorporate a lower-triangular matrix $\boldsymbol{L} \in \mathbb{R}^{N \times N}$ into the covariance model to capture aspects of covariance that are condition-independent (i.e. do not depend on $\boldsymbol{x}$):

$$\mathbf{\Sigma}(\boldsymbol{x}) = \boldsymbol{L}(\boldsymbol{U}(\boldsymbol{x})\boldsymbol{U}(\boldsymbol{x})^\top + \mathbf{\Lambda}(\boldsymbol{x}))\boldsymbol{L}^\top, \tag{8}$$

In the literature $\boldsymbol{L}\boldsymbol{L}^T$ is referred to as the scale matrix. Some intuition can be gained by assuming $\mathbf{\Lambda}(\boldsymbol{x}) = \mathbf{0}$ and $P > N$, in which case $\boldsymbol{U}(\boldsymbol{x})\boldsymbol{U}(\boldsymbol{x})^\top \propto \boldsymbol{I}$ in expectation under the marginal prior distribution. Thus, a good choice is to set $\boldsymbol{L}$ to be the Cholesky factor of the grand empirical covariance as defined in eq. (2). In our experiments, we use this as an initialization for $\boldsymbol{L}$ and we optimize it to maximize the evidence lower bound (ELBO) as described in the Supplement B.3.

It is also possible, and potentially beneficial, to instead model smoothness in the inverse covariance matrix (also known as the precision matrix). This results in an *inverse Wishart process*. We refer the reader to Supplement B.2 for additional details of the inverse Wishart model.

## 2.4 Latent Variable and Parameter Inference

Each of the probabilistic models above specifies a posterior distribution of the noise covariance conditioned on observed neural activity, $p\left(\{\mathbf{\Sigma}(\boldsymbol{x}_c)\}_{c=1}^{C} \mid \{\boldsymbol{y}_{ck}\}_{c=1,k=1}^{C,K}\right)$. More precisely, each model specifies a posterior on the latent covariance factor matrices, $p\left(\{\boldsymbol{U}(\boldsymbol{x}_c)\}_{c=1}^{C} \mid \{\boldsymbol{y}_{ck}\}_{c=1,k=1}^{C,K}\right)$, which defines the desired posterior over covariance through a measurable transformation (eq. 8). This distribution over latent covariance factors has $NPC$ dimensions, precluding us from representing the posterior exactly (doing so would require us to tile $\mathbb{R}^{NPC}$ with a lattice of points). We instead use mean field variational inference [48] to approximate the desired posterior. Since this approach has already been described for similar Wishart process models [49], we relegate a description to Supplement B.3. All codes are implemented in `numpyro` [50] and available at `https://github.com/neurostatslab/wishart-process`.

## 2.5 Connections to Population Decoding

As we discussed in section 1, the estimation of noise covariance matrices is an important step for a variety of downstream neural analyses. Here, we briefly highlight two applications of Wishart process models in the context of neural decoding and perceptual discrimination [3].

**Quadratic Discriminant Analysis (QDA)** A simple decoding task is to infer which condition $c \in \{1, \ldots, C\}$ gave rise to an observed neural response, $\boldsymbol{y} \in \mathbb{R}^N$, on a single trial. A natural approach is to use a maximum likelihood classifier, $\texttt{class}(\boldsymbol{y}) = \arg\max_{c \in \{1, \ldots, C\}} \log p(\boldsymbol{y}|\hat{\boldsymbol{\mu}}_c, \hat{\mathbf{\Sigma}}_c)$. Doing so requires that we construct estimates for the mean responses, $\hat{\boldsymbol{\mu}}_1, \ldots, \hat{\boldsymbol{\mu}}_C$, and covariances, $\hat{\mathbf{\Sigma}}_1, \ldots, \hat{\mathbf{\Sigma}}_C$. Since covariance estimation is challenging, a common practice (see e.g. [10]) is to use the grand empirical covariance (eq. 2) and therefore set $\hat{\mathbf{\Sigma}}_1 = \cdots = \hat{\mathbf{\Sigma}}_C$. Under this model, known as *linear discriminant analysis* (LDA; see e.g. [51] Ch. 4), it can be shown that the decision boundaries of the maximum likelihood classifier become linear.

Intuitively, if we can accurately estimate distinct covariances for each condition, we may unlock more powerful decoders by relaxing the assumption of equal covariance across conditions. Under this model, known as *quadratic discriminant analysis* (QDA), the decision boundaries are described by hyperbolas (A schematic is shown in Fig. 2D). Furthermore, Pagan, Simoncelli, and Rust [52] described a circuit implementation of QDA using a cascade of two linear-nonlinear models, suggesting that such classifiers are a simple and biologically plausible neural computation. We demonstrate the ability of Wishart process models to enable accurate decoding via QDA in Fig. 2E.

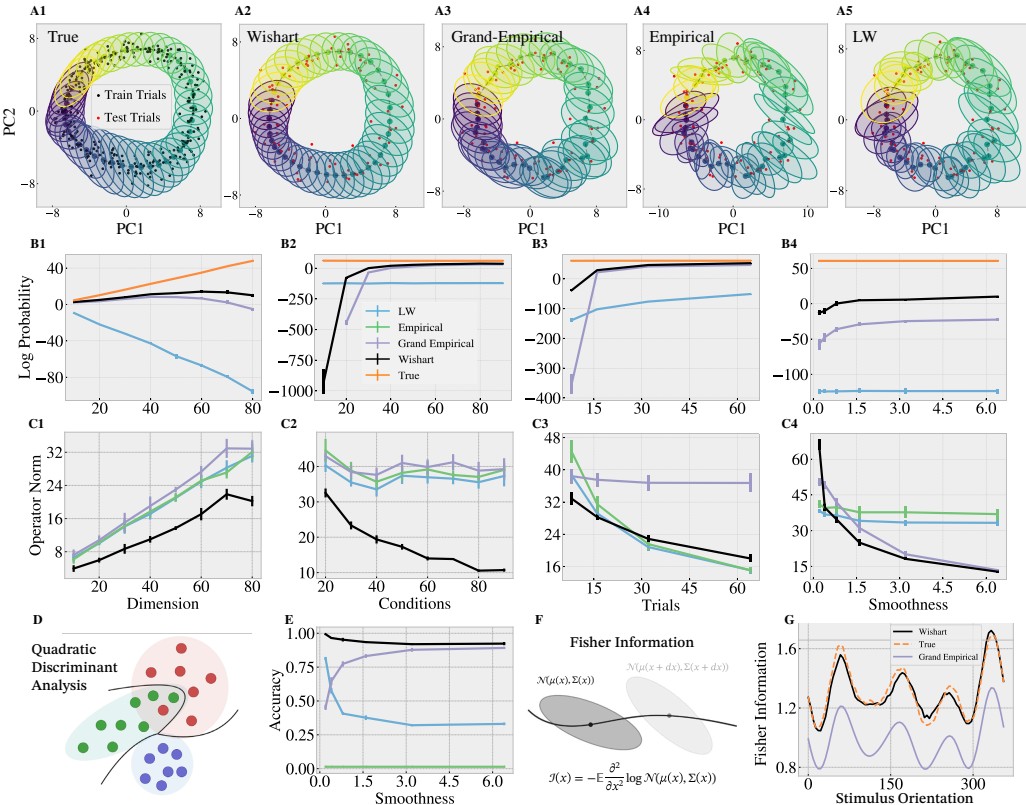

**Figure 2:** Results on synthetic data. See Supplement B.1 for all simulation details. (A) Simulated (A1) and inferred (A2-A5) response distributions using different methods. Means $\boldsymbol{\mu}(\boldsymbol{x}_c)$ and covariances $\boldsymbol{\Sigma}(\boldsymbol{x}_c)$ are shown in dashed lines and colored ellipses in PC space. Train and test trials $\boldsymbol{y}_{ck}$ are shown in black and red dots. (B1-B4) Mean log-likelihood of held-out trials as a function of the number of dimensions, conditions, trials, and covariance smoothness. Error bars represent median $\pm$ SEM across 5 independent runs. (C1-C4) Same as in B, the operator norm between the true and inferred covariances. (D) Schematic of QDA classifier. (E) Using different estimates of covariance, we use QDA to infer the condition label of held-out trials. Error bars represent the standard deviation over 6 independent runs. (F) Schematic of Fisher Information, FI. (G) Estimation of FI using grand empirical covariances and Wishart model.

**Fisher Information (FI)** A quantity that plays an important role in our understanding of how neural noise properties affect behavioral discriminability and representation is FI defined as $\mathbb{E}[\partial^2 \log p(\boldsymbol{x})/\partial \boldsymbol{x}^2]$ [3]. Unfortunately, estimating FI is challenging because it involves measuring infinitesimal changes in the neural response distribution with respect to the stimulus condition (Fig. 2F). Wishart processes enable us to exploit the continuous parametrization of experimental conditions to derive a natural estimator for FI. For constant covariances across conditions, the FI for multivariate normal distributions is given by $\boldsymbol{\mu}'(\boldsymbol{x})^T \boldsymbol{\Sigma}^{-1} \boldsymbol{\mu}'(\boldsymbol{x})$ where $\boldsymbol{\mu}'(\boldsymbol{x})$ is the gradient of the mean with respect to $\boldsymbol{x}$. If covariances vary across conditions, this quantity is replaced with $\boldsymbol{\mu}'(\boldsymbol{x})^T \boldsymbol{\Sigma}^{-1}(\boldsymbol{x}) \boldsymbol{\mu}'(\boldsymbol{x}) + \frac{1}{2}\texttt{tr}([\boldsymbol{\Sigma}^{-1}(\boldsymbol{x})\boldsymbol{\Sigma}'(\boldsymbol{x})]^2)$ where $\boldsymbol{\Sigma}'(\boldsymbol{x})$ is the gradient of the covariance w.r.t. the conditions and $\texttt{tr}$ is the trace operator. Wishart process models enable the continuous estimation of FI as a function of $\boldsymbol{x}$. This is achieved by sampling from the gradient of the inferred posterior distribution (see Supplement B.6).

## 3 Results

We qualitatively and quantitatively compared Wishart processes to common covariance estimation methods used in neural data analysis: Ledoit-Wolf [19], empirical covariance (eq. 1), grand empirical covariance (eq. 2), and graphical LASSO [21]. Our primary measure of performance is the log-

likelihood that the models assign to held-out data. Because the baseline methods are not Bayesian, we handicap ourselves by using a single sample from the posterior distribution to calculate held-out log-likelihoods. We also visualize and qualitatively evaluate contours of the predictive probability density ("covariance ellipses") in the top two principal component subspace ("PC space"). For synthetic data, we report the difference between inferred and true covariances measured in the operator norm (Fig. 2C1-C4). We observed that generally log-likelihood and operator norm results are aligned. However, since they measure the performance in two different ways it is possible for a model to outperform another in log probability but vice versa on the covariance operator norm.

## 3.1 Synthetic Data

We first validate our approach on simulated data over a periodic 1-dimensional space, with conditions sampled at $C$ equispaced angles on the interval $[0, 2\pi)$. The mean and covariance of neural responses were sampled according to the Wishart process generative model (eq. 4) and trials for each condition were conditionally sampled according to eq. (3) from a multivariate Gaussian distribution. An example of generated data is shown in Fig. 2A1. Due to our choice for the condition space, we use periodic kernels both for the Gaussian and Wishart process priors. For simplicity, we set $\mathbf{\Lambda}(\boldsymbol{x})$ to the identity matrix both in the data generative and inference models. To match experimental observations [47], we simulated low-dimensional noise covariance structure by setting scale matrix to $\boldsymbol{L}\boldsymbol{L}^\top = \boldsymbol{U}\mathtt{diag}(s_{1:N})\boldsymbol{U}^T$ where $\boldsymbol{U}$ is a uniformly sampled orthogonal matrix in $N$ dimensions and $s_1 > s_2 > \cdots > s_N > 0$ are logarithmically spaced numbers between 1 and $10^{-5}$. Except for the number of dimensions, conditions, trials, and Wishart smoothness ($\lambda_\Sigma$), which we vary over a range to assess the performance of different models, all other hyperparameters are kept constant ($P = 2$, $\gamma = 0.001$, $\beta = 1$, and $\lambda_\mu = 1$). Supplement B.1 provides further details for these simulations, covering all specific hyperparameter settings for each panel of Fig. 2.

We fit Wishart process models to these simulated datasets using the procedures described in section 2.4. These models qualitatively capture the ground truth covariance structure (Fig. 2A2-A5) and quantitatively outperform alternative methods across a range of synthetic datasets. Wishart process inference performs particularly well in high-dimensional regimes relevant to neural data analysis with large $N$, small $K$, and intermediate levels of smoothness across conditions (Fig. 2B-C). We also investigated the ability of Wishart process models to improve decoding accuracy on this dataset via QDA, as summarized in section 2.5. The Wishart process model outperformed the baseline models across a range of data smoothness suggesting that it captures the boundaries between classes more accurately (Fig. 2E). Furthermore, in Fig. 2G we show that on synthetic data the Wishart process model accurately estimates the FI continuously across conditions.

## 3.2 Primate Reaching

To study how the brain plans upcoming actions, animals are often trained to prepare voluntary movements to cued locations during a "delay period" before executing those movements after a "go cue" is delivered [53–55]. During this delay period, neural firing rates in the primary motor cortex (M1) and dorsal premotor cortex (PMd) correlate with features of the upcoming movement (e.g. direction and distance) in nonhuman primates [56, 57]. Trial-to-trial variability in the delay period has been suggested to correlate with fluctuations in reaction time [58], but a quantitative account of this variability with dense target configurations is challenging to achieve [29].

We analyzed 200ms of spiking activity prior to the onset of the "go cue" across $N = 192$ multiunit channels from both PMd and M1 in a nonhuman primate executing point-to-point hand reaches.[3] The dataset contains $C = 48$ conditions, where targets were organized in three concentric circles at radii of 4cm, 8cm, and 12cm, with 16 equidistant targets around each ring (Fig. 3A). We used a squared exponential kernel (eq. 5) to model smoothness across reaches of different lengths and a $2\pi$-periodic kernel (eq. 6) to model smoothness across reach angles. Models were fitted on 60% of the available trials, radial and angular smoothness parameters for both GP and WP kernels were selected on a validation set of 25% of the data, and log-likelihood scores were generated using the remaining 15% of trials. This procedure was repeated over 8 randomized data partitions.

The Wishart process model outperforms competing estimators (Fig. 3B). Since $N > K$, the empirical covariance (eq. 1) is singular and therefore produces negative infinity log-likelihoods. Instead,

---

[3]Data was previously published and made available by [29].

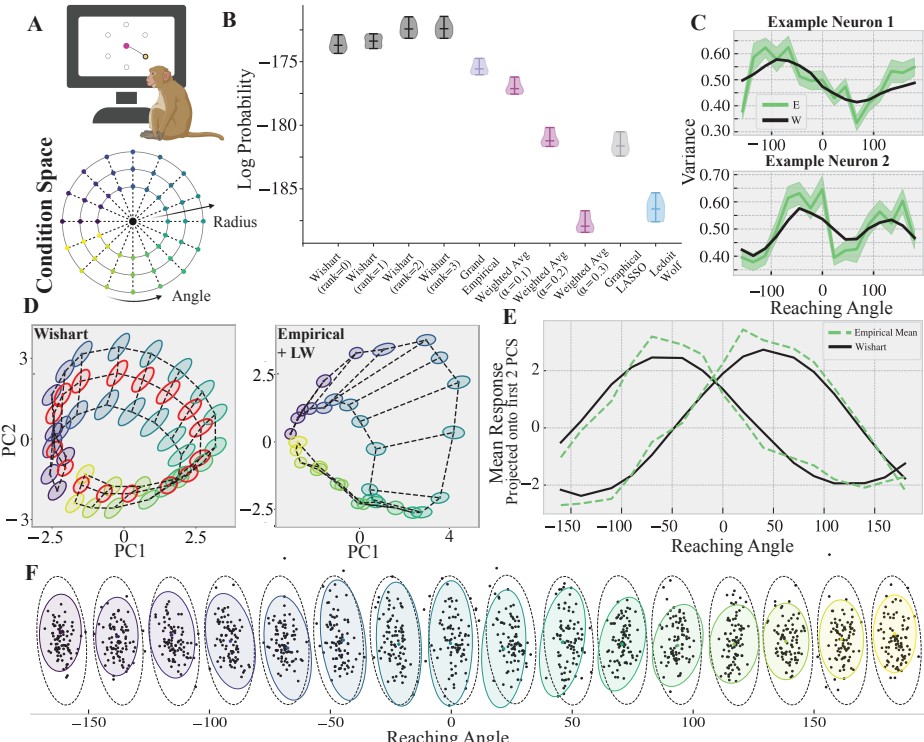

**Figure 3:** Results on primate reaching task. (A) Task schematic (*left*) and set of experimental conditions (*right*). (B) Log-likelihood of held-out trials across 8 randomized folds. Horizontal lines indicate the maximum, minimum, and mean of the log-likelihood distribution. (C) Marginal variance of two example neurons across reaching angles, averaged over reach radii (green) and Wishart process model estimate (black). Shaded green denotes mean $\pm$ SEM. (D) Wishart (*left*) and Ledoit-Wolf (*right*) estimates of mean and covariance, projected onto the first two principal components. To demonstrate the ability of the Wishart Process model to predict covariance in unseen conditions, the middle ring of targets was held out in training (red ellipses). (E) Predicted mean and (F) covariance in held-out conditions shown in panel D. Covariance ellipses, grand-empirical estimates (dashed lines), and samples (black dots) are visualized in the top-2 principal component subspace.

weighted averages of the grand empirical and empirical covariance were calculated according to $\mathbf{\Sigma}^{\mathrm{WA}}(\boldsymbol{x}_c) = \alpha\mathbf{\Sigma}^{\mathrm{E}}(\boldsymbol{x}_c) + (1 - \alpha)\mathbf{\Sigma}^{\mathrm{GE}}$ (WA, E, and GE denote weighted average, empirical, and grand-empirical estimators respectively). This baseline is a simplified variant of [20] shrinking the per-condition covariances towards the grand-empirical estimate. Performance results produced by this baseline are shown in Fig. 3B at different values of $\alpha$.

These results suggest that WP is able to capture the smooth change in the covariance across conditions. Since smooth changes in the covariance imply smooth changes in the marginal variance (diagonal terms in the covariance), we plot the marginal variance of individual neurons as a function of the reaching angle to visualize the smooth change. We see that the variance indeed changes across conditions and that the Wishart process model provides a smooth fit to these noisy variance estimates (Fig. 3C). This smoothness is absent in the predictions of the Ledoit-Wolf and graphical LASSO estimators, while the grand covariance estimate is constant and, in essence, over-smoothed (Supp. Fig. 2). The out-of-sample log probability results suggest that the smoothness captured by WP is indeed a property that is present in the data and has physiological relevance (Fig. 3B).

We also investigated the Wishart model's ability to interpolate covariance estimates for unseen conditions. Here, the model was retrained on only trials from the inner and outermost rings (4cm and 12cm). The neural response means and covariances around the center ring were then estimated by sampling from the model posterior evaluated at these hold-out condition locations (8cm, all angles).

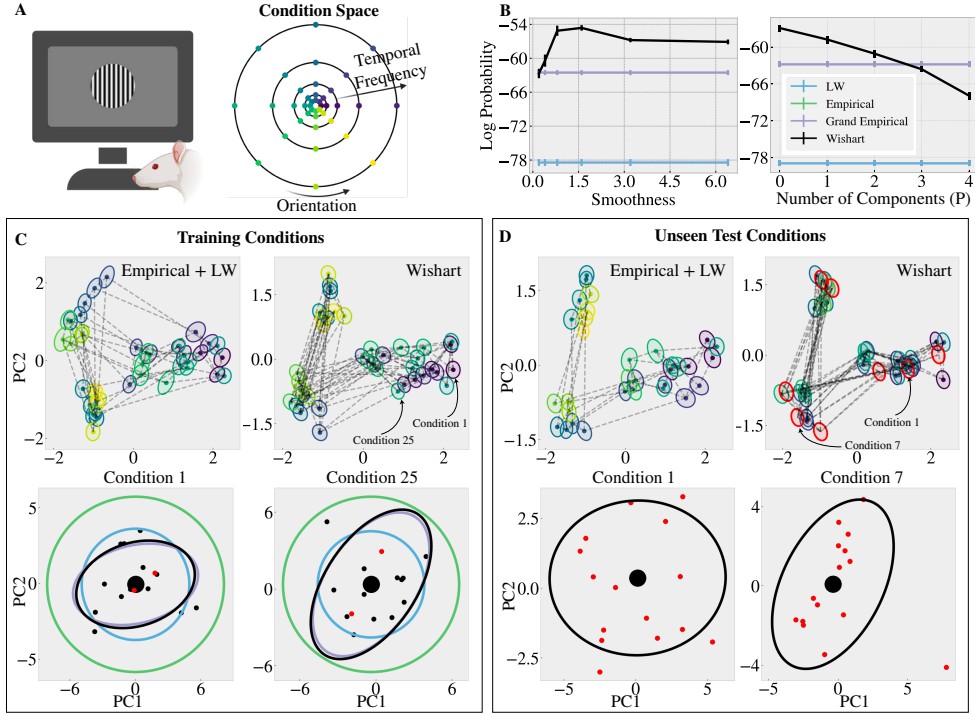

**Figure 4:** Results on drifted gratings dataset. (A) Task schematic (*left*) and set of experimental conditions (*right*). (B) Cross-validated log probability as a function of Wishart smoothness ($\lambda_\Sigma$) and the number of components ($P$) achieving their optimal values at $\lambda_\Sigma \approx 1.5$ and $P = 0$ and outperforming alternatives. (C) Top: mean responses and noise covariances of the full set of conditions plotted in the PC space for the Ledoit-Wolf (*left*) vs. Wishart (*right*) estimators (covariances are scaled down by a factor of 100 for visualization purposes). Colors are in the increasing order of condition number matching the colors in (A); Bottom: Train and test trials (black and red dots) in two example conditions plotted in the top two PC subspace. Covariance ellipses are color-coded by the model as in panel B. (D) Generalizing to unseen conditions: Top: 80% of the conditions are used for training (shown in colors) and the remaining conditions (shown in red) are withheld. Bottom: Inferred covariances by Wishart Process model for two held-out conditions (as in panel C, bottom). Notice that panels C, and D are from two separate analyses. In panel C, we train on all conditions and test on held out trials. In panel D, we train on a subset of conditions and test on held out conditions (interpolation over stimulus space). Condition 1 was held out only in the second analysis.

Wishart model estimations perform comparably to empirical mean and grand-empirical covariance estimations calculated on these hold-out conditions (Fig. 3E-F).

### 3.3 Allen Brain Observatory (Neuropixels)

Perceptual discrimination is a longstanding area of research in psychology and neuroscience, dating back to work in the early 1800s by Weber and Fechner [59]. A classic paradigm requires subjects to discriminate between pairs of oriented visual gratings, which humans and nonhuman primates accomplish with high precision [60–62]. Discrimination thresholds on this task are thought to be related to the scale and orientation of noise in neural representations [7] (Fig. 1A), and recent works have examined this hypothesis by simultaneously recording large populations of neurons in rodent models [10, 25, 26]. The resulting analyses are delicate and require a large number of trials.

To investigate, we analyzed simultaneously recorded responses in mouse primary visual cortex to drifting visual gratings in the Allen Brain Observatory (Visual Coding: Neuropixels dataset).[4] We analyzed a session (session-id = 756029989) with 81 neurons. Spike count responses in a 200 ms

---

[4]https://portal.brain-map.org/explore/circuits/visual-coding-neuropixels

window from the stimulus onset were recorded across 4 grating orientations, 2 directions, and 5 temporal frequencies as illustrated in Fig. 4A. We used periodic kernels for the orientation and direction axes (eq. 6), and a squared exponential kernel (eq. 5) for the temporal frequency axis of the condition space. For each of the resulting 40 conditions, 15 trials are collected, which we split into 13 train vs. 2 test trials. For choosing the number of components ($P$) and the smoothness of the Wishart kernel ($\lambda_\Sigma$), we quantified the cross-validated log probability of test trials over 10 randomized folds. As depicted in Fig. 4B the optimal performance of the Wishart model is achieved for $\lambda_\Sigma \approx 1.5$ and $P = 0$.

This selection of the model parameters allows us to capture the structure that is shared among all conditions (note the similarity between Wishart and grand empirical covariance in Fig. 4C), while representing the condition-specific changes in the covariance structure (note the smooth change of matrix orientation across conditions in the top-right plot of Fig. 4C). Finally, we present covariance interpolation results across unseen conditions where we fit the model according to 32 randomly selected train conditions (colored ellipses in Fig 4D top-left) and leave out the complete set of trials for the remaining 8 conditions (red ellipses in Fig 4D top-right). The interpolated means and covariances for two example test conditions present a strong agreement with their corresponding held-out trials (Fig 4D bottom row).

# 4  Conclusion

We proposed a method to estimate covariance across large populations of co-recorded neurons, which is widely regarded as a challenging problem [8, 9] with important implications for theories of neural coding [3, 5, 7]. Specifically, we formulated probabilistic Gaussian and Wishart process models [42] to model smoothly changing covariance structure across continuously parameterized experimental conditions. We found this approach outperforms off-the-shelf covariance estimators that do not exploit this smoothness (e.g. Ledoit-Wolf) as well as the grand covariance estimator (eq. 2) which is infinitely smooth (i.e. does not allow for condition-dependent covariance). We investigated two task modalities (vision and motor) and model organisms (mouse and nonhuman primate). However, we expect these modeling principles will be even more broadly applicable, including to other sensory modalities with smoothy parameterized stimuli (e.g. acoustic tones or odorant mixtures) and in neural circuits supporting navigation [30, 31].

We believe our work also provides an important conceptual advance. Given the limited duration of experiments, it may seem that there is a strict tradeoff between accurately estimating neural responses across many experimental conditions and estimating trial-by-trial noise statistics in any fixed condition. Our results show that this tradeoff is not so simple. Indeed, if neural responses are sufficiently smooth across conditions, the Wishart process model could predict neural covariance accurately by densely sampling conditions with only a single trial per condition (e.g. visual gratings at random orientations, as done in [25]). Intuitively, the model works by pooling information across nearby conditions, and so can even predict covariance in entirely unseen conditions (as demonstrated in Figs. 3D and 4D). Thus, the limiting factor in estimating noise covariance is not the number of trials per condition, but the smoothness of the neural responses over the space of experimental conditions.

We conclude by discussing the current limitations of our model and opportunities for future work. First, the model requires us to explicitly specify kernel functions ($k_\mu$, $k_\Sigma$) over the space of experimental conditions. We studied simple paradigms where a reasonable parametric form of this kernel could be specified and the hyperparameters could be fine-tuned through cross-validation. Extending the model to more complex stimulus sets (e.g. to naturalistic image stimuli) would be of great interest, and likely require more sophisticated approaches for learning kernels and feature representations of these stimulus sets. Second, we assumed that trial-to-trial noise followed a Gaussian distribution. A deeper investigation of noise models, including multivariate extensions of Poisson distributions [39, 47, 63], is warranted. Finally, we utilized a basic variational inference procedure to approximate the posterior distribution over neural covariance. While fully representing the posterior is intractable (as is typically the case), more sophisticated optimization techniques may further benefit performance [64]. Nonetheless, our results show that Wishart process models are performant even in the absence of the aforementioned extensions. These models outperform common practices in neuroscience and open the door to new analyses, such as the prediction of noise covariance in held-out conditions.

## Acknowledgements

We thank Saurabh Vyas (Columbia) and Eric Trautmann (Columbia) for helpful discussions and for providing data from the nonhuman primate motor cortex. We also thank Sam Zheng (NYU) and Sarah Harvey (NYU) for their feedback on the manuscript.

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
