# SUPPLEMENTARY MATERIAL
## Estimating Noise Correlations Across Continuous Conditions With Wishart Processes

## A  Supplemental Figures

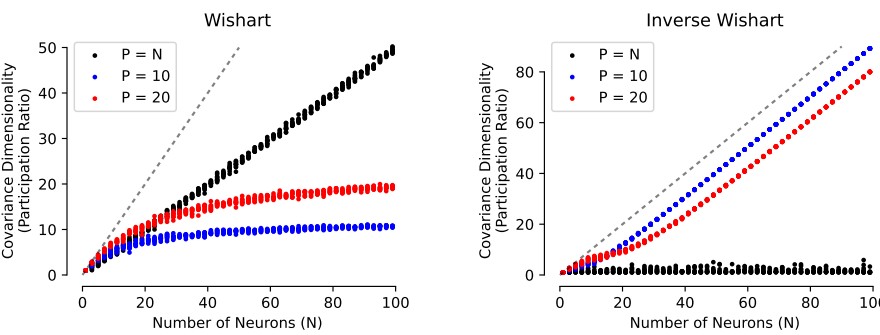

**Figure 5:** The dimensionality of covariance matrices sampled from the prior distribution under Wishart and Inverse Wishart models (black) as a function of the number of neurons ($N$). The dimensionality of the covariance matrix is quantified by the participation ratio: $\sum_i (\lambda_i)^2 / \sum_i \lambda_i^2$, where $\lambda_i$ are the eigenvalues of the matrix. *Left*, Wishart distribution prior, $\boldsymbol{\Sigma} = \boldsymbol{U}\boldsymbol{U} + \sigma^2 \boldsymbol{I}$ where the elements of $\boldsymbol{U} \in \mathbb{R}^{N \times P}$ are standard normal variables $\mathcal{N}(0, 1)$. For the full rank model ($P = N$ in black), $\sigma^2 = 0$. For low-rank models ($P = 10$ and $P = 20$ in blue and red), $\sigma^2 = 0.1$. Dashed black line is the identity line. Results for 10 random samples from the prior for each neural population size (different values of $N$). *Right*, same as *left* but with Inverse Wishart distribution prior, $\boldsymbol{\Sigma} = \left(\boldsymbol{U}\boldsymbol{U} + \sigma^2 \boldsymbol{I}\right)^{-1}$ where $\boldsymbol{U}$ is a random Gaussian matrix as described above.

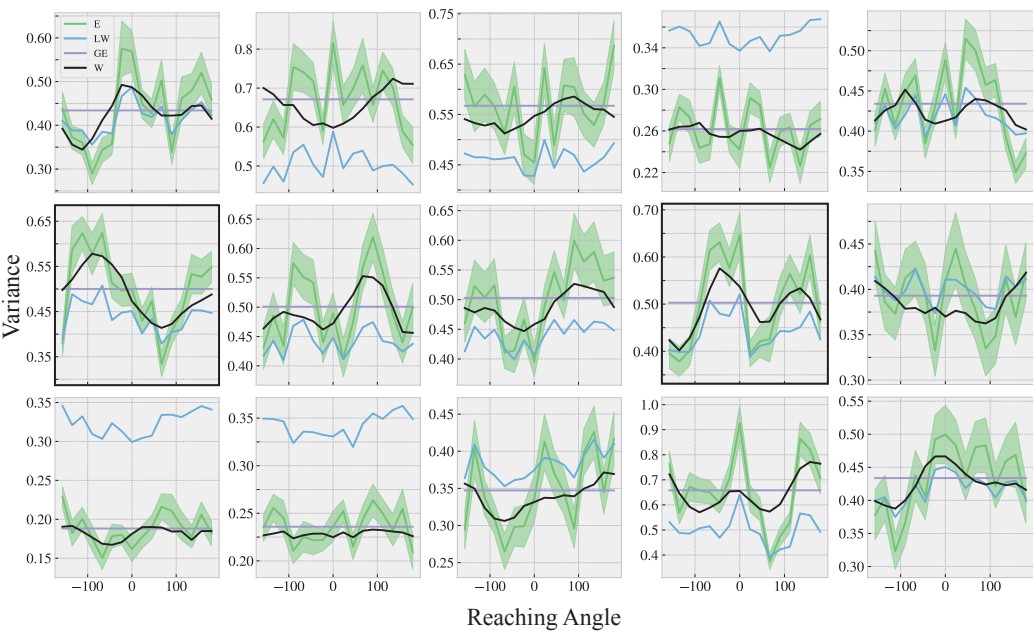

**Figure 6:** Marginal variance estimations of 12 example neurons across reaching angles. Wishart (*black*), Ledoit-wolf (*blue*), grand-empirical (*purple*), and empirical (*green*) marginal variance estimates. Graphical LASSO uses empirical variance to estimate the diagonal and is therefore equivalent to the empirical estimate (*green line*) in this depiction [1].

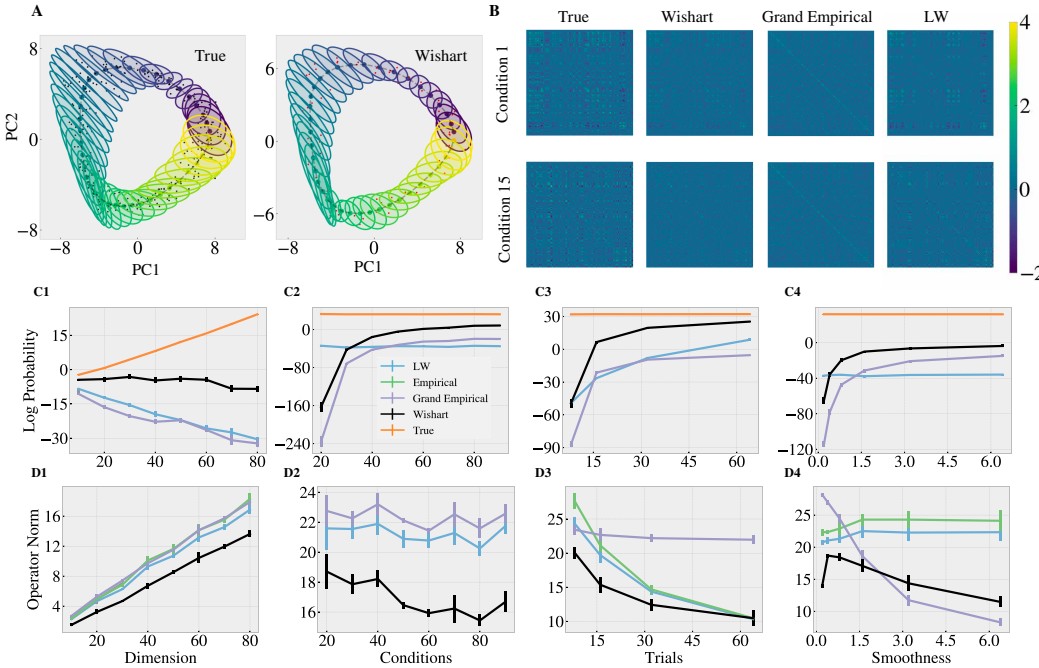

**Figure 7:** Results on synthetic data for $\boldsymbol{LL}^T = \boldsymbol{I}$. (A) Similar setting to Fig. 2, with parameters $\{N, C, K, \lambda_\Sigma, P\} = \{100, 40, 10, 1.0, 4\}$. Means $\boldsymbol{\mu}(\boldsymbol{x}_c)$ and covariances $\boldsymbol{\Sigma}(\boldsymbol{x}_c)$ are shown in dashed lines and colored ellipses respectively for the ground truth (*left*) vs. Wishart (*right*) plotted in the PC space. Train and test trials $\boldsymbol{y}_{ck}$ are shown in small black and red dots. (B) Covariances inferred by different methods for two example conditions ($\boldsymbol{x}_1 = 0°$, $\boldsymbol{x}_{15} = 168°$). Wishart captures the low-rank structure observed in the true matrices using a small number of trials per condition. (C1-C4) Log probability on held-out trials as a function of the number of dimensions (10-90), conditions (10-90), trials (8,16,32,64), and covariance smoothness (0.2, 0.4, 0.8, 1.6, 3.2, 6.4). For each plot, we vary one parameter and fix other parameters to $\{N, C, K, \lambda_\Sigma, P\} = \{100, 40, 10, 1.0, 4\}$. Error bars represent median $\pm$ SEM across 5 independent runs for each parameter configuration. (D1-D4) Mean operator norm between the true and inferred covariances showing agreement with the log probability results.

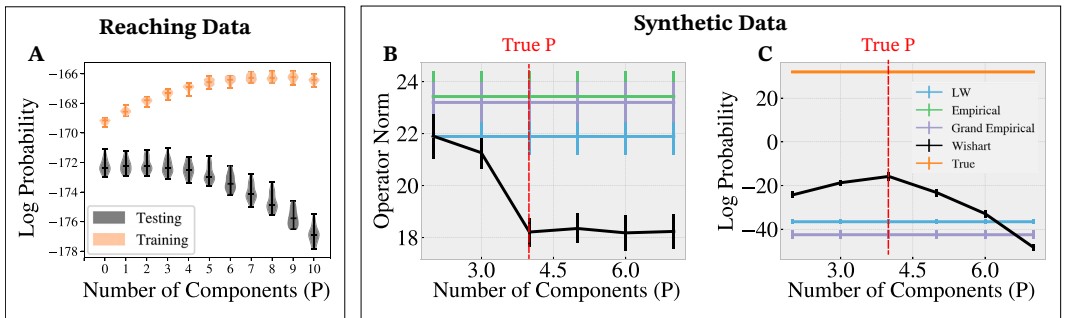

**Figure 8:** Validation of the number of components ($P$). (A) Log probabilities of training and hold-out trials on 3-ring stimulus arrangement of the primate reaching task, across a range of components. Probability on the held-out set begins to decline after $P = 2$, while the probability of the training set continues to increase, evidence of overfitting to the training set at higher $P$. (B-C) Results on synthetic data generated from ground truth model with $P_{\text{true}} = 4$; Wishart model fitted with varying $P$ (shown on the x-axis) achieving the lowest operator norm error (B) and the highest likelihood (C) when $P$ coincides $P_{\text{true}}$.

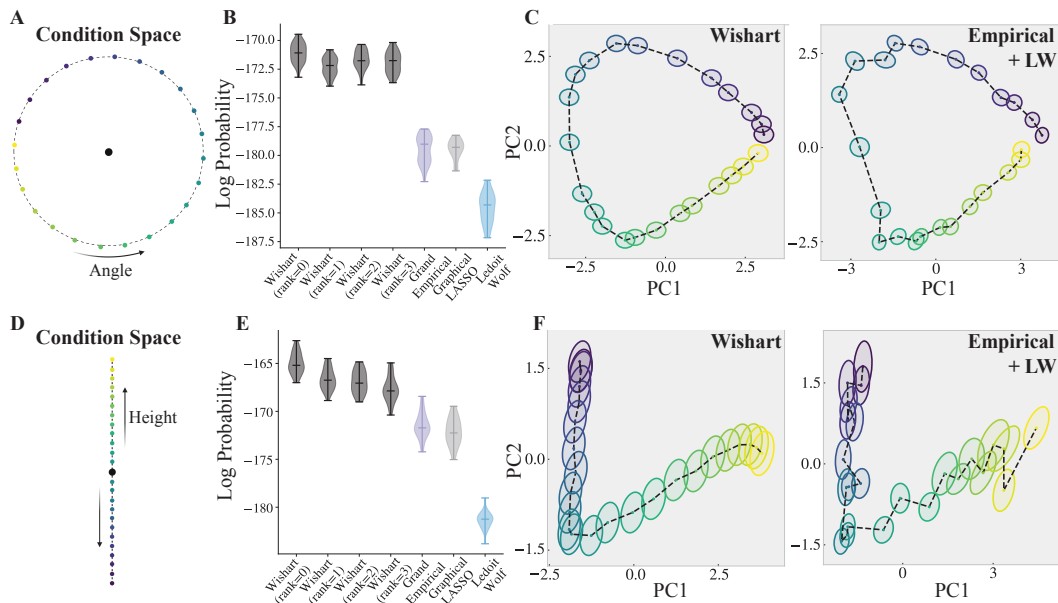

**Figure 9:** Results on additional target configurations of primate reaching task. (A-C) Reaching targets are arranged on a single ring with 24 targets. This dataset contains 60 trials per reaching condition - 80% were used as training trials, and the remaining 20% for testing. A $2\pi$-periodic kernel was used to model smoothness around the ring. Scale parameters were identical to those used in Fig. 3, as was the angular smoothness parameter. (A) Reaching targets are arranged equidistantly around a ring with a radius of 8cm. (B) Log probability of hold-out trials, over 30 random folds. (C) Wishart (*left*) and Ledoit-Wolf (*right*) estimates of mean and covariance projected onto the first two principal components. (D-F) Reaching targets are arranged on a vertical line with 24 targets. The dataset contains 70 total trials per condition, which are similarly divided into 80% training and 20% testing trials. We used a squared exponential kernel to model smoothness along the line. Scale parameters were again identical to the ones in Fig. 3. (D) Reaching targets are equally spaced 1cm apart along the vertical line. (E,F) are the same as (B,C) for the linear arrangement. Our results are in line with [2] Figure 2A.

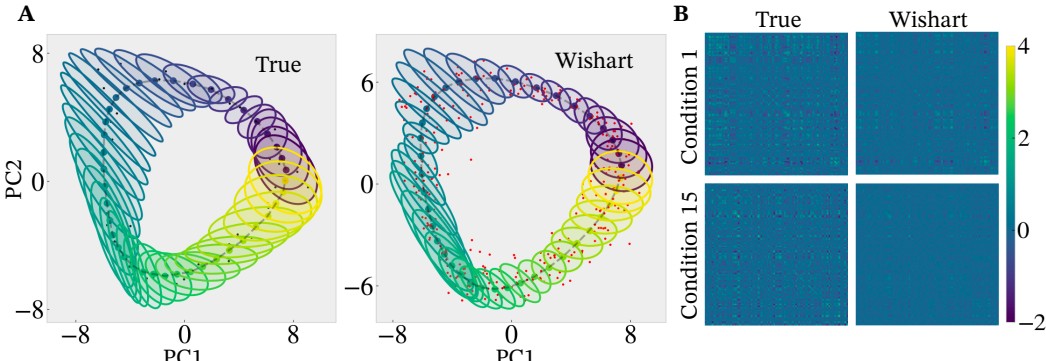

**Figure 10:** Inferring covariances using a single trial per condition. (A) Left: Synthetic data generated from the ground truth model, with a single training trial per condition (black dots). Right: Covariances inferred by the Wishart model overlaid on the test trials (red dots). (B) True and inferred covariances for two example conditions (1, 15). Wishart recovers the coarse structure of the true covariances using a single trial per condition.

# B  Supplemental Methods

## B.1  Simulation Details

Data used in Fig. 2A1-A5 was simulated according to the generative model in equation 4 in 100 dimensions. The parameters used in this simulation are as follows: 100 neurons ($N = 100$), 40 conditions ($C = 40$), 10 trials per condition split into 8 training and 2 testing trials per condition ($K = 8$). The mean covariance matrix $L$ was generated by exponentially decaying eigenvalues of rank 2 to mimic the low-dimensionality of neural responses. The condition space was parameterized by a periodic value ranging from 0-360 (akin to reach angle or visual grating orientation) and hence periodic kernel was used for this analysis. We set the number of components to 2 ($P = 2$) for both the generative and inference models. The diagonal scaling of the kernel is set to 0.001 ($\gamma_{\mu}, \gamma_{\Sigma} = 0.001$) for both mean GP and covariance WP kernels. We added a diagonal matrix of 0.1 times identity to all per-condition covariance matrices to ensure they are PSD. The smoothness parameter for both GP and WP was set to 1 ($\lambda_{\mu}, \lambda_{\Sigma} = 1$). We used multivariate Gaussian conditional likelihood and drew IID samples to generate train and test trials for each condition. We used GP mean estimation in 2A2 and empirical mean estimation for 2A3-A5.

We performed variational inference using mean field normal distribution as the approximating family. The number of particles for importance weighting was set to 1. We performed SGD optimization using `Adam` optimizer with the step size of 0.001 using 50,000 iterations without mini-batching. In all experiments, we initialized the mean of the variational family according to the per-condition empirical means of the data and initialized the inference scale matrix according to the Cholesky factorization of the grand-empirical covariance.

For other panels in Fig. 2 we used a similar scheme to the one described above. However, in every panel, we changed one parameter of the generative model and kept all the other parameters constant. We varied the number of dimensions between 10 and 90 with steps of 10, number of conditions between 20 and 100 with steps of 10, number of trials per condition in $\{8, 16, 32, 64\}$ and the smoothness parameter for the covariance in $\{.2, .4, .8.1.6, 3.2, 6.4\}$. The fixed parameters in each plot were chosen as $N = 100, C = 30, N = 10, \lambda_{\Sigma} = 1$. In all the log probability plots (Fig. 2B) we used empirical mean estimation to compare the performance of covariance estimators in isolation. All the results in Fig. 2 are based on the held-out test trials.

## B.2  Inverse Wishart Process

The inverse Wishart process is defined as follows:

$$\Sigma(\boldsymbol{x}) = \left( \boldsymbol{L} \left( \boldsymbol{U}(\boldsymbol{x}) \boldsymbol{U}(\boldsymbol{x})^{\top} + \boldsymbol{\Lambda}(\boldsymbol{x}) \right) \boldsymbol{L}^{\top} \right)^{-1}. \tag{1}$$

Since the mapping $\Sigma \mapsto \Sigma^{-1}$ is continuous over (strictly) positive definite matrices, imposing smoothness on $\Sigma^{-1}(\boldsymbol{x})$ implies some degree of smoothness in $\Sigma(\boldsymbol{x})$. Nonetheless, the Wishart and inverse-Wishart process specify distinct prior distributions, and it is an empirical question of which one is better suited to any specific circumstance (see Fig. 5 for the dimensionality of covariance matrices sampled from the priors of Wishart and inverse-Wishart models). Importantly, one can perform inference in eq. (1) without representing and inverting the covariance matrix explicitly, which would lead to numerical instabilities. Therefore, eq. (1) may be preferable when the inverse covariance is of greater interest than the covariance—e.g., in calculations of linear Fisher information [3].

## B.3  Inference

The inference algorithm that we propose is similar to [4], except that we do not utilize inducing points as the number of conditions in neuroscience experiments is often small. Instead, we directly perform variational inference on the latent variables $\{\boldsymbol{U}(\boldsymbol{x}_c)\}_{c=1}^{C}$ and $\{\boldsymbol{\mu}(\boldsymbol{x}_c)\}_{c=1}^{C}$. This allows us to easily switch between MAP estimates of the variables and posterior inference by choosing delta or normal distribution as approximating variational family with the latter introducing further regularization.

We denote the generative parameters by $\boldsymbol{\theta}$, which in our case only contains the $\boldsymbol{L}$ matrix. Given the choice of $q_{\boldsymbol{\phi}}$, we would like to find $q_{\boldsymbol{\phi}}\left( \{\boldsymbol{U}(\boldsymbol{x}_c), \boldsymbol{\mu}(\boldsymbol{x}_c)\}_{c=1}^{C} \right)$ to best approximate $p_{\boldsymbol{\theta}}\left( \{\boldsymbol{U}(\boldsymbol{x}_c), \boldsymbol{\mu}(\boldsymbol{x}_c)\}_{c=1}^{C} \mid \{\boldsymbol{y}_{ck}\}_{c=1,k=1}^{C,K} \right)$ while optimizing over the parameters $\boldsymbol{\theta}$. We follow the

literature on stochastic variational inference and derive the ELBO cost function for this problem.

$$\mathcal{L}(\boldsymbol{\phi}, \boldsymbol{\theta}) = \mathbb{E}_q\big[\log p_{\boldsymbol{\theta}}\left(\{\boldsymbol{U}(\boldsymbol{x}_c), \boldsymbol{\mu}(\boldsymbol{x}_c)\}_{c=1}^{C}, \{\boldsymbol{y}_{ck}\}_{c=1,k=1}^{C,K}\right) - \log q_{\boldsymbol{\phi}}\left(\{\boldsymbol{U}(\boldsymbol{x}_c), \boldsymbol{\mu}(\boldsymbol{x}_c)\}_{c=1}^{C}\right)\big]$$

Using the reparameterization trick, we achieve an empirical estimate of the gradient of loss using samples from an independent noise.

$$\boldsymbol{z} \coloneqq \left(\{\boldsymbol{U}(\boldsymbol{x}_c)\}_{c=1}^{C}, \{\boldsymbol{\mu}(\boldsymbol{x}_c)\}_{c=1}^{C}\right) \quad \boldsymbol{z} = h_{\boldsymbol{\phi}}(\boldsymbol{\epsilon})$$

$$\nabla_{\boldsymbol{\phi}, \boldsymbol{\theta}}\mathcal{L}(\boldsymbol{\phi}, \boldsymbol{\theta}) = \mathbb{E}_{\boldsymbol{\epsilon}}\big[\nabla_{\boldsymbol{\phi}, \boldsymbol{\theta}}\log p_{\boldsymbol{\theta}}\left(\{\boldsymbol{x}_c\}_{c=1}^{C}, \{\boldsymbol{y}_{ck}\}_{c=1,k=1}^{C,K}, h_{\boldsymbol{\phi}}(\boldsymbol{\epsilon})\right) - \nabla_{\boldsymbol{\phi}}\log q_{\boldsymbol{\epsilon}}(h_{\boldsymbol{\phi}}(\boldsymbol{\epsilon}))\big]$$

In practice, we use Monte Carlo estimates of the expectation above by sampling from an independent noise distribution and evaluating the gradient inside the expectation for those samples. The number of samples used for estimating the gradient introduces a trade-off between the accuracy of the estimation computation time. We empirically find that a single sample is often enough to achieve accurate estimates. We choose `Adam` optimizer with the step size $0.001$ to perform our stochastic gradient-based optimization.

### B.3.1 Computational Complexity

In order to examine the computational complexity of our variational framework, we first expand the gradient calculation. Notice that the complexity of our model is not the same as out-of-the-box stochastic variational models, due to the specific structure in our prior model, namely Gaussian and Wishart Processes.

$$\nabla_{\boldsymbol{\phi}, \boldsymbol{\theta}}\mathcal{L}(\boldsymbol{\phi}, \boldsymbol{\theta}) = \mathbb{E}_{\boldsymbol{\epsilon}}\big[\nabla_{\boldsymbol{\phi}, \boldsymbol{\theta}}\log p_{\boldsymbol{\theta}}\left(\{\boldsymbol{x}_c\}_{c=1}^{C}, \{\boldsymbol{y}_{ck}\}_{c=1,k=1}^{C,K}, h_{\boldsymbol{\phi}}(\boldsymbol{\epsilon})\right) - \nabla_{\boldsymbol{\phi}}\log q_{\boldsymbol{\epsilon}}(h_{\boldsymbol{\phi}}(\boldsymbol{\epsilon}))\big]$$

$$\approx \sum_{c,k}\nabla_{\boldsymbol{\phi}, \boldsymbol{\theta}}\log p_{\boldsymbol{\theta}}\left(\boldsymbol{x}_c, \boldsymbol{y}_{ck} \mid h_{\boldsymbol{\phi}}(\tilde{\boldsymbol{\epsilon}})\right) + \nabla_{\boldsymbol{\phi}, \boldsymbol{\theta}}\log p_{\boldsymbol{\theta}}(h_{\boldsymbol{\phi}}(\tilde{\boldsymbol{\epsilon}})) - \nabla_{\boldsymbol{\phi}}\log\mathcal{N}(h_{\boldsymbol{\phi}}(\tilde{\boldsymbol{\epsilon}}); \boldsymbol{0}, \mathbb{1})$$

The first term (log observation), requires inverting and the determinant of the per-condition covariance matrices, which has a complexity of $\mathcal{O}(CN^3)$ and $K$ matrix multiplications with the covariance matrices which has a complexity of $\mathcal{O}(CKN^2)$. The second term (log prior) requires inverting $P$ matrices that are of size $C \times C$ and, therefore have the complexity of $\mathcal{O}(C^3P)$. Notice that if the kernel parameters are not optimized, the kernel matrix only needs to be evaluated once, not for every iteration.

Overall, the complexity is given by $\mathcal{O}(CN^3 + CKN^2 + C^3P)$, which can be improved by mini-batching (turning the multiplicative term $CK$ into the batch size). Furthermore, if we assume that we have few conditions $C \sim \mathcal{O}(1)$ and the number of neurons scales quadratically with the number of trials $K \sim \mathcal{O}(\sqrt{N})$ and use mini-batching this reduces the overall complexity to $\mathcal{O}(N^3)$. Notice that these computations are required for a single iteration of SVI.

We observed that on GPU it takes about 80 seconds to fit a dataset with 100 neurons, 80 conditions, and 32 trials per condition. Notice that we run 10000 iterations of our optimization algorithm, therefore the run time for such a dataset is about 8 milliseconds per iteration. The runtime for other algorithms for a similarly sized dataset are the following (in seconds): PoSCE: 45, Graphical Lasso: 24, Ledoit-Wolf: 0.3, and Empirical: 0.3. All other algorithms were run on CPU as GPU implementations are not available.

Notice that our model has fundamentally new capabilities that are not present in simple baselines. In particular, we can infer a continuous manifold for the mean and covariance of neural responses (see Fisher Information analysis), generalize to entirely unseen conditions, and quantify our uncertainty in a Bayesian framework. A fair comparison between inference times should take this difference into account.

### B.4 Posterior Predictive Distribution

Posterior predictive distributions provide a principled approach in our model to sample means and covariances in training and unseen test conditions. Posterior predictive defines a distribution over unseen test pair $(\boldsymbol{x}^*, \boldsymbol{y}^*)$ conditioned on the training data with the inferred posterior marginalized out. In practice, posterior predictive likelihoods can be estimated using samples from the approximate posterior distribution in the following way.

$$p_{\hat{\boldsymbol{\theta}}}(\boldsymbol{x}^*, \boldsymbol{y}^* | \mathcal{D}) = \int p_{\hat{\boldsymbol{\theta}}}(\boldsymbol{x}^*, \boldsymbol{y}^* | \boldsymbol{z}) p_{\hat{\boldsymbol{\theta}}}(\boldsymbol{z} | \mathcal{D}) d\boldsymbol{z} = \mathbb{E}_{p_{\hat{\boldsymbol{\theta}}}(\boldsymbol{z} | \mathcal{D})}[p_{\hat{\boldsymbol{\theta}}}(\boldsymbol{x}^*, \boldsymbol{y}^* | \boldsymbol{z})]$$

$$\approx \frac{1}{S} \sum_{\boldsymbol{z}_s \sim p_{\hat{\boldsymbol{\theta}}}(\boldsymbol{z} | \mathcal{D})} p_{\hat{\boldsymbol{\theta}}}(\boldsymbol{x}^*, \boldsymbol{y}^* | \boldsymbol{z}_s) \approx \frac{1}{S} \sum_{\boldsymbol{z}_s \sim q_{\hat{\boldsymbol{\phi}}}(\boldsymbol{z})} p_{\hat{\boldsymbol{\theta}}}(\boldsymbol{x}^*, \boldsymbol{y}^* | \boldsymbol{z}_s)$$

where every $\boldsymbol{z}_s$ for $s = \{1, \ldots, S\}$ are $S$ samples from the inferred posterior on $\left(\{\boldsymbol{U}(\boldsymbol{x}_c)\}_{c=1}^C, \{\boldsymbol{\mu}(\boldsymbol{x}_c)\}_{c=1}^C\right)$ and $p_{\hat{\boldsymbol{\theta}}}(\boldsymbol{x}^*, \boldsymbol{y}^* | \boldsymbol{z}_s)$ is the model likelihood conditioned on the sampled $\boldsymbol{z}_s$. If $\boldsymbol{x}^*$ is among the training conditions, this likelihood is a straightforward evaluation of a multivariate Gaussian with parameters given by $\boldsymbol{U}(\boldsymbol{x}^* = \boldsymbol{x}_c), \boldsymbol{\mu}(\boldsymbol{x}^* = \boldsymbol{x}_c)$. If the test condition is not observed among the training conditions, we can apply the Bayes rule again and estimate its likelihood. Notice that this interpolation property is allowed in our model due to the Gaussian and Wishart process priors. Sampling from the model and evaluating its likelihood for unobserved conditions is not possible in previous models since the models render different conditions independent and do not consider a smoothness prior.

$$p\left(\boldsymbol{x}^*, \boldsymbol{y}^* \mid \{\boldsymbol{U}(\boldsymbol{x}_c), \boldsymbol{\mu}(\boldsymbol{x}_c)\}_{c=1}^C\right) = \mathbb{E}_{p\left(\boldsymbol{U}(\boldsymbol{x}^*), \boldsymbol{\mu}(\boldsymbol{x}^*) \mid \{\boldsymbol{U}(\boldsymbol{x}_c), \boldsymbol{\mu}(\boldsymbol{x}_c)\}_{c=1}^C\right)}[p(\boldsymbol{y}^* \mid \boldsymbol{U}(\boldsymbol{x}^*), \boldsymbol{\mu}(\boldsymbol{x}^*))]$$

where samples from $p\left(\boldsymbol{U}(\boldsymbol{x}^*), \boldsymbol{\mu}(\boldsymbol{x}^*) \mid \{\boldsymbol{U}(\boldsymbol{x}_c), \boldsymbol{\mu}(\boldsymbol{x}_c)\}_{c=1}^C\right)$ are drawn from a multivariate Gaussian with a covariance provided by the kernel.

### B.5 Spiking Count Model

It has been argued that for a dataset of spike counts a gain-modulated multivariate Poisson likelihood model better represents first and second-order statistics of the data while providing flexibility for incorporating and estimating neural correlations [5].

$$\boldsymbol{\mu}(\boldsymbol{x}) \sim \mathcal{GP}^N \quad \boldsymbol{\Sigma}(\boldsymbol{x}) \sim \mathcal{WP}^{lrd}(\boldsymbol{L}, P)$$

$$\boldsymbol{g}_{ck} | \boldsymbol{x}_c \sim \mathcal{N}(\boldsymbol{g} | \boldsymbol{\mu}_c, \boldsymbol{\Sigma}_c) \quad \boldsymbol{y}_{ck} | \boldsymbol{g}_{ck} \sim \text{Poiss}(\boldsymbol{y} | \texttt{softplus}(\boldsymbol{r} + \boldsymbol{g}_{ck}))$$

where $\boldsymbol{g}_{ck}$ is a vector of gains sampled from a multivariate normal distribution, $\boldsymbol{y}_{ck}$ is the vector of spike counts for condition $c$, and trial $k$, and $\boldsymbol{r}$ is the vector of condition-independent baseline spike rates for individual neurons that we optimize. The graphical model of this statistical model is shown in Fig. 1F.

**Inference Details** We highlight the differences between running inference in the Normal and Poisson models. The **latent variables** of the Poisson model include $\boldsymbol{g}_{ck}$ and our mean field distribution factorizes over $\boldsymbol{\mu}_{1:C}, \boldsymbol{\Sigma}_{1:C}, \boldsymbol{g}_{1:C,1:K}$. Hence, our posterior is defined over the joint space of these variables. The **parameters** of the generative model in addition to the prior parameters (i.e. $\boldsymbol{L}$) include the likelihood rate parameters as well $\boldsymbol{\theta} = \{\boldsymbol{L}, \boldsymbol{r}\}$. As before, we optimize the ELBO cost function and run the stochastic variational inference with `Adam` optimizer. We observed that a larger **step size** (0.005) and more **optimization iterations** (50000) are often needed to achieve the best results.

**Summary Statistics** In the Poisson model the means and covariances are defined and estimated in the gain space instead of the spike count space. However, our statistical model allows for sampling the means and covariances in the spike count space using a simple Monte Carlo scheme.

$$\mathbb{E}ss(\boldsymbol{y}) \approx \frac{1}{S} \sum_{\substack{\boldsymbol{\mu}_s, \boldsymbol{\Sigma}_s \sim q_{\hat{\boldsymbol{\phi}}}(\boldsymbol{\mu}, \boldsymbol{\Sigma} | \mathcal{D}) \\ \boldsymbol{g}_s \sim p(\boldsymbol{g} | \boldsymbol{\mu}_s, \boldsymbol{\Sigma}_s) \\ \boldsymbol{y}_s \sim p_{\hat{\boldsymbol{\theta}}}(\boldsymbol{y} | \boldsymbol{g}_s)}} ss(\boldsymbol{y}_s)$$

for any summary statistic function $ss$. If we choose $ss(\boldsymbol{y}) = \boldsymbol{y}$, the equation above will return the estimate of mean spike counts. If we choose $ss(\boldsymbol{y}) = (\boldsymbol{y} - \mathbb{E}\boldsymbol{y})(\boldsymbol{y} - \mathbb{E}\boldsymbol{y})^T$ it will return the estimate of the covariance of the spike counts.

**Likelihood Evaluation** To compare which of the Poisson or Normal models is a better fit to the data, we need to compare the corresponding likelihoods. However, not having access to the gain

parameter in the Poisson model makes the exact likelihood evaluation intractable. Instead, we will marginalize out the gain parameter using the following integral similar to the posterior predictive.

$$p(\boldsymbol{y}|\mathcal{D}) = \int p_{\hat{\boldsymbol{\theta}}}(\boldsymbol{y}|\boldsymbol{g})p(\boldsymbol{g}|\boldsymbol{\mu}, \boldsymbol{\Sigma})p_{\hat{\boldsymbol{\theta}}}(\boldsymbol{\mu}, \boldsymbol{\Sigma}|\mathcal{D})d\boldsymbol{g}d\boldsymbol{\mu}d\boldsymbol{\Sigma}$$

$$\approx \int p_{\hat{\boldsymbol{\theta}}}(\boldsymbol{y}|\boldsymbol{g})p(\boldsymbol{g}|\boldsymbol{\mu}, \boldsymbol{\Sigma})q_{\hat{\boldsymbol{\phi}}}(\boldsymbol{\mu}, \boldsymbol{\Sigma}|\mathcal{D})d\boldsymbol{g}d\boldsymbol{\mu}d\boldsymbol{\Sigma}$$

which in practice is performed using Monte Carlo samples from the corresponding distributions. A comparison of the log-likelihood values achieved by the Poisson and Normal models for Poisson-generated data is shown in Fig. 11.

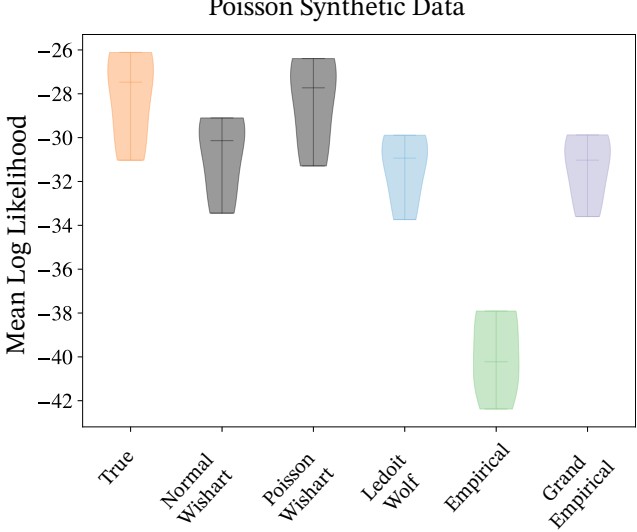

**Figure 11:** Results from synthetically generated Poisson firing rates. We generated data from the model in eq. B.5 with 20 neurons, 30 conditions, and 50 trials per condition. We then inferred means and covariances separately using the Normal and Poisson models and evaluated the test log-likelihoods of the held-out trials. The Poisson model produced log-likelihood values closest to the ground truth and outperformed the Normal model as well as other compared methods.

### B.6 Fisher Information Estimation

Suppose $f : \mathbb{R}^M \to \mathbb{R}$ is distributed according to the Gaussian Process $\mathcal{GP}(f; \mu, K)$ where $\mu, K$ are mean and kernel functions, and assuming that $\frac{\partial \mu}{\partial x_i}(\boldsymbol{x})$ and $\frac{\partial^2 K}{\partial x_i \partial x_i'}(\boldsymbol{x}, \boldsymbol{x})$ exist for all $i$ and any $\boldsymbol{x}, \boldsymbol{x}'$ then for $f$ and its gradient $f'$ we have the following result thanks to the linearity of differentiation operation.

$$p(f, \nabla f) = \mathcal{GP}\left( \begin{bmatrix} f \\ f' \end{bmatrix}; \begin{bmatrix} \mu \\ \mu' \end{bmatrix}, \begin{bmatrix} K & \frac{\partial K}{\partial \boldsymbol{x}}(\boldsymbol{x}', \boldsymbol{x})^T \\ \frac{\partial K}{\partial \boldsymbol{x}}(\boldsymbol{x}, \boldsymbol{x}') & \frac{\partial^2 K}{\partial \boldsymbol{x} \partial \boldsymbol{x}'}(\boldsymbol{x}, \boldsymbol{x}') \end{bmatrix} \right)$$

Since the posterior GP is another GP with data-dependent mean and kernel functions, we can jointly sample a function and its gradient or compute the gradient of the posterior mode.

Given this result, we can sample from the posterior derivative and compute the uncertainty associated with it. In addition, we can compute derived quantities such as Fisher Information as described in the main text.