# OpenReview forum: "Estimating Noise Correlations Across Continuous Conditions With Wishart Processes"
_NeurIPS.cc/2023/Conference — NeurIPS 2023 poster_

### Official Review · Reviewer_ruJu · 2023-06-22

**Soundness:** 2 fair
**Presentation:** 4 excellent
**Contribution:** 3 good
**Rating:** 7
**Confidence:** 4

**Summary:**

The goal of this paper is to compute the covariance of recorded neurons in a given stimulus condition with a low number of samples. Although the conditions are different, some aspects are shared which justifies the fact that the covariance for a specific condition should depend on the covariance in other conditions. This idea is implemented using a Bayesian model which imposes smoothness of the means and models the covariance as $L (U U^\top + \Lambda) L^\top$ where $L$ does not depend on the condition and $U$ and $\Lambda$ evolve smoothly with the condition.

**Strengths:**

Overall, the paper is very well-written and addresses a significant problem in neuroscience.
The use of smoothness in the means and covariance estimates across conditions is new as far as I know. The authors validate their approach by experimenting with synthetic and real data.


**Weaknesses:**

While the smoothness part is new, exploiting similar conditions to yield a better covariance estimate is not. The example I have in mind is [1] where condition-specific covariances are biased towards the population covariance but since this idea is very natural, I assume more examples exist.

[1] Rahim, Mehdi, Bertrand Thirion, and Gaël Varoquaux. "Population shrinkage of covariance (PoSCE) for better individual brain functional-connectivity estimation." Medical image analysis 54 (2019): 138-148.

In particular, I find the experiments a bit biased toward the success of the method presented by the authors. The absence of any competitor that uses both empirical and population covariance estimates makes the comparison a bit unfair (obviously models that do not do this will not perform well by design of the generative mechanism).

The other major drawback is the lack of clear metrics of performance on real data that demonstrate predictive power (e.g. a regression or classification task). The current metrics on the real data experiment rely mostly on visual inspection. As a side note, the visualization of principal components may hide some of the signal especially if the generated data is non-linear. Predictive tasks are better in that they give a clearer metric of performance. Is there a way to predict something related to the subjects (their age, their performance for instance) so that we have a proxy for the quality of the estimated covariance?

A potential weakness (although I am not sure for this one), is the lack of internal cross-validation. Unless I am wrong, I believe the hyper-parameters are optimized on the same cross-validation loop as the one used for testing. If I am wrong, please clarify in the text by clearly stating that there are two nested cross-validation loops (an internal CV on the train set and an outer one to select the train and test splits).

Lastly, compared to Ledoit-Wolf or others, I would assume that the model presented by the author is much slower to fit. If this is the case, this should be clearly stated (a figure with the running time in function of N would be great).

There are small parts in the submission that I did not understand:
- How do you optimize L?
- Is $\bar{\mu}$ in equation (4) the average over samples in all conditions ? I believe this is never defined.
- When you compute the log-likelihood (for instance to produce Figure 3 B) you need a model of the data. Is this model the one described by (3) ?



**Questions:**

- Can you comment on the absence of a method mixing population and condition-specific covariance ?
- Is there a way to design a predictive task based on your data instead of relying on visual inspection?
- Can you comment on the computation time of your method ?
- How is the cross-validation made? Is there an internal cross-validation to select the hyper-parameters?

**Limitations:**

N.A

---

> ### Author Rebuttal · Authors · 2023-08-09
>
> > **While the smoothness part is new, exploiting similar conditions [...].**
>
> Thank you for the pointer to the PoSCE paper. We had overlooked it since it is a neighboring field (none of us have experience with fMRI data).
>
> Roughly speaking, PoSCE can be adapted to our problem by estimating the covariance in condition $c$ as a weighted combination of the empirical/sample covariance in condition $c$ and the grand covariance across conditions. Our original work already compared to the two extremes – sample covariance and grand covariance were included as baselines – but it is a good idea to include a weighted combination of these two estimates as a stronger baseline.
>
> In actuality, PoSCE posits a probabilistic model over the tangent space of the PSD manifold and uses a geometric mean to perform the averaging. However, when we applied their code to our data it spits out NaNs. Given time constraints, and since conceptually simpler baselines may be preferable to a reader anyways, we decided to take a weighted linear combination of the per-condition sample covariance and the grand covariance. We feel this is equally justified as the Riemannian distance used by PoSCE, since sample covariances are already the arithmetic mean of the second moments: $\Sigma = (1/n)\sum_i \mathbf{x_i x_i^\top}$.
>
> We find that the Wishart process outperforms this modified PoSCE baseline on real and synthetic data (see “Weighted Avg” results in Rebuttal Fig F).
>
> Importantly, even if the PoSCE baseline was competitive in terms of performance, we still think our work would stand as an important contribution. In particular, PoSCE is not able to capture smoothness across conditions (as the reviewer mentions). It therefore cannot extrapolate or interpolate between conditions, which is one of our central motivations. See, e.g., the application to Fisher Information in the general rebuttal.
>
> > **The other major drawback is the lack of clear metrics [...].**
>
> Thank you for this suggestion. Inspired by this comment, we’ve performed a new analysis to demonstrate how improved covariance estimation may lead to improved decoding of the experimental condition. See Rebuttal Fig C and our general response to all reviewers for more details about this experiment.
>
> > **A potential weakness [...] is the lack of internal cross-validation [...].**
>
> Our original submission used an initial cross-validation run to show how sensitive the log-likelihood was to hyperparameters such as smoothness and number of components in Fig. 4B. We then fixed these hyperparameters and ran them over 30 additional train-test folds. Since each of these 30 folds was a fully randomized split, we think it is unlikely that we would have overfit our small number of hyperparameters.
>
> However, we agree with the reviewer that the most rigorous way to do this is to have randomized cross-validation train-validation-test folds, and to select new hyperparameters for each fold on the basis of the validation set. We have now done this on the monkey data and show our results in Rebuttal Fig F. The results are effectively unchanged, but we will swap this new analysis into the final revision because we agree that it is more rigorous.
>
> We will also clearly explain the cross-validation process in our methods – in particular, for each train-validation-test fold we do a randomized hyperparameter search over the kernel smoothness parameters and select the best model on the validation set for testing. We welcome any additional requests for details.
>
> > **Lastly, [...] the model presented by the author is much slower to fit [...].**
>
> Please check our response to R1 for the wall clock time comparisons. Although the computational complexity of the model is tractable (as mentioned in supplementary B.1.1) our model takes longer than compared methods to run. Notice that this is due to a multitude of reasons. First, instead of a single estimate of the covariance, we infer a whole distribution over means and covariances per condition which can be used to compute uncertainties associated with inferred means and covariances and also to sample from them. In addition, since our model is GP, WP based, we uncover an underlying continuous space of means and covariances which can be used for interpolating test conditions and computing properties such as curvature and gradients wrt conditions. Developing full-blown Bayesian statistical models like these usually require more computational resources as opposed to ad-hoc point-estimate or frequentist counterparts. Although our implementation is relatively fast, we recognize that there’s still a lot of room for improving the run time of the algorithm using the latest developments in the GP literature. We leave this for future work.
>
> > **How do you optimize L?**
>
> The matrix $L$ is initialized according to the Cholesky factorization of the grand empirical covariance and it’s optimized to maximize the evidence lower bound (ELBO). Note that $L$ is not a latent random variable: it is a parameter that we optimize as a point estimate. In contrast, we optimize a distribution over the latent variables of the model (e.g. covariance factors, denoted $U$ in the paper). These are jointly optimized with parameters like $L$. We regret that our description was too brief and caused confusion; we will improve the revision to make the paper more self-contained.
>
> > **Is $\bar{\mu}$ in equation (4) [...].**
>
> We apologize for this oversight. $\bar{\mu}$ is the mean function for the prior GP, in all of our experiments we set it to the constant function zero, we will include both of these missing details in the revised manuscript.
>
> > **When you compute the log-likelihood [...].**
>
> Yes. In all of our original analyses F(.) is a multivariate normal distribution as described on line 104. Since our original submission, we have also extended the model to be compatible with Poisson noise (see Rebuttal Fig E).
>
> All the questions in the **Questions** section are addressed above.

---

> > ### Comment · Reviewer_ruJu · 2023-08-16
> >
> > Thanks for a comprehensive rebuttal. I believe this paper should be accepted. I have increased my score to 7.

---

### Official Review · Reviewer_yodS · 2023-06-29

**Soundness:** 2 fair
**Presentation:** 3 good
**Contribution:** 2 fair
**Rating:** 5
**Confidence:** 4

**Summary:**

This work proposes to use Wishart Processes, originally proposed by Wilson and Ghahramani, to estimate the covariance structure of neural activity in experiments where there are parametric variations linking stimulus conditions. By pooling estimates appropriately across task conditions, the limited number of experimental trials can be used to provide more reliable estimates of variability. Inference is performed via a mean field variational approach, and the model is applied to synthetic data and two neural data sets.

This seems like a reasonable approach to data of the type the authors are modeling, and the experiments show that it outperforms existing methods in the literature. Conversely, this is a very straightforward application of an existing method, which may be novel to neuroscientists, but does not represent much of new conceptual contribution. Moreover, the results presented, while likely better estimates of covariance, do not produce any neuroscientific insights reported here.

**Strengths:**

- Application of Bayesian nonparametric techniques for estimating covariance to a new domain.
- Pools strength across parametrically related stimulus conditions, making good use of a limited number of trials.
- Clear presentation.

**Weaknesses:**

- This is a relatively straightforward application of an existing method.
- Inferential details of the model are somewhat sparsely described. (See questions below.)
- Results for the experiments are evaluated on various performance metrics, but it is unclear what new insights this method provides. It is perfectly reasonable to employ a better estimation method for this model, but the experiments in Figures 3 and 4 do not produce any additional findings beyond evidence that the new method works better.

**Questions:**

- ll. 26-29: The number of parameters grows quadratically, but if all pairs of neurons are simultaneously observed, it doesn't follow that the process of estimating correlations does not scale well. Rather, the argument should be from the limited number of trials in each condition, right?
- What is the difference between a Wishart process and a GP with a kernel that includes both trial and neuron indices? Some discussion of this might be helpful for those unfamiliar with the method.
- Details of the variational approximation in the supplement are somewhat sparse. Do I understand correctly that the authors are performing full GP inference on $\mathbf{U}$ and $\boldsymbol{\mu}$? What exactly is the mean field form of $q_\phi$? Does it factorize over $(\mathbf{U}, \boldsymbol{\mu})$ or something else? Finally, the reference on line 7 of Supplement B is not included there, and I don't think it's the same as reference 3 in the main text.

**Limitations:**

- This work is focused on neuroscience tasks in which trials are sampled from a set of parametrically related experimental conditions. This is true of many kinds of experiments but will not apply to many other cases that do not share that structure.
- The authors discuss (in Supplement B) several methods for mitigating the poor scaling behavior of Gaussian Processes with $N$, $P$, and $C$ but do not implement any of the well-known scalable GP methods that would mitigate this, since their examples are for only moderate $N$.

---

> ### Author Rebuttal · Authors · 2023-08-09
>
> > ***This is a relatively straightforward application of an existing method***
>
> This is one of the most important points to discuss so we have laid out a detailed response in our “general rebuttal." To summarize our arguments:
> * “Application” papers of all stripes are requested in the “NeurIPS Call for Papers”
> * Our submission fits into a specific tradition of method papers in the “Neuroscience and Cognitive Science” track.
> * We extended the model to incorporate low-rank plus diagonal structure and we have now further extended the model to handle Poisson-distributed noise.
> * Even “vanilla” Wishart Processes are relatively exotic models that (to our knowledge) have not been widely applied outside of quantitative finance.
>
> > ***The number of parameters grows quadratically, but if all pairs of neurons are simultaneously observed, it doesn't follow that the process of estimating correlations does not scale well***
>
> Our original statement&mdash;"number of estimated parameters grows quadratically... while the number of measurements grows only linearly"&mdash;is correct.
>
> E.g. Imagine a case where `N=2` neurons and `K=150` trials. Thus, we have `N * K = 300` measurements of firing rate. We want to estimate a `2 x 2` symmetric covariance, which has 3 parameters. Thus, the ratio of measurements to parameters is 100 to 1.
>
> Now imagine if we have `N=100` neurons, and we want to estimate `N * (N + 1) / 2` parameters in the covariance matrix. Intuitively, if we want the ratio of measurements to parameters to stay the same as before (i.e. 100 to 1), we would need to chose `K` to satisfy `N * K = 100 * N * (N + 1) / 2`. That is, we would need `K = 100 * 201 / 2 = 10050` trials.
>
> In summary, the difficulty of the problem depends *both* on `N` and `K`. We will edit our paper to convey this more clearly. Note that the intuition we provided above can be made fully rigorous. E.g. Vershynin (2011) “How Close is the Sample Covariance Matrix to the Actual Covariance Matrix?”
>
> > ***What is the difference between a Wishart process and a GP with a kernel that includes both trial and neuron indices?***
>
> As a refresher, our method uses a GP to estimate the mean response and a Wishart process (WP) to estimate the covariance. Both the GP and WP have a kernel that measures similarity *across different experimental conditions*. The hyperparameters of this kernel are related to how smooth the neural response is as a function of changing the condition. In our model there is no kernel that measures similarity across neurons or trials. We consider these extensions in turn below:
>
> *GP model with kernel over neurons.* This would incorporate a smoothness prior over the neuron indices (e.g. neuron 1’s response would be highly correlated with neuron 2’s response, and de-correlated with neuron 53’s response). In most experiments, neurons are labeled arbitrarily so it typically does not make sense to add this structure.
>
> *GP model with kernel over trial.* This would incorporate a smoothness prior over trial index, which could be helpful to model slow drift or non-stationarity in the neural response. Since multiple conditions are often randomly interleaved across trials, it may be better to use the absolute time between two trials (rather than their integer index) to quantify this non-stationarity.
>
> Note that adding a time component to the GP kernel  *would only model non-stationarity in the mean* neural response.* It would *not* capture correlation structure across neurons (which is our primary motivation). Thus, an interesting extension of our model would be to add a time component to *both the GP and WP kernels*.
>
> We welcome further discussion from the reviewer, in case we have misinterpreted their suggestion.
>
> > ***it is unclear what new insights this method provides***
>
> We agree that we can do more to spell out the scientific insights our model helps enable. We would like to point the reviewer to some existing results in the paper, as well as some new experiments that we’ve performed in the rebuttal phase.
>
> *Results in the paper:* (a) smooth covariance interpolation in Fig. 3D  (b) Generating covariances using a single trial per condition in Supplemental Fig. 6 (c) drawing a full posterior distribution over means and covariances attaching uncertainties to covariance estimates.
>
> *New results:* (a) assessing Fisher Information in Rebuttal Fig 1D (b) modeling covariances for Poisson distributed observations in Rebuttal Fig 1E (c)  linear vs. quadratic discriminant analysis in Rebuttal Fig 1C.
>
> Finally, the introduction section of our paper cites multiple neuroscience papers where covariance plays an important quantitative role. Our model is broadly applicable, and so opens up the possibility of many new scientific insights across these cited works.
>
> We hope that the reviewer can understand it is challenging to validate a new form of analysis while simultaneously providing a scientific breakthrough. The primary focus of our paper is methodological, but we have done our best to outline scientific future directions (see our "general rebuttal").
>
> > ***Details of the variational approximation in the supplement are somewhat sparse.***
>
> We regret that we were not clear and understand the importance of fixing these details. By “full GP inference” we interpret the reviewer to mean the closed form solution to GP regression with Gaussian observation noise. We do not use this anywhere since the WP posterior does not admit a closed form solution. Thus, to approximate the posterior we perform joint variational inference over the GP and WP parameters (they do not factorize as pointed out by Reviewer mr49).  Each parameter is modeled with a Gaussian with a learnable mean and variance, and no correlations are modeled across parameters. That is, we perform standard mean field variational inference (e.g. Blei et al. 2017, “Variational Inference: A Review for Statisticians”).
>
> We are happy to answer more queries in the discussion period.

---

> > ### Comment · Reviewer_yodS · 2023-08-10
> >
> > I appreciate the authors responses to my review. Some specific replies:
> >
> > > “Application” papers of all stripes are requested in the “NeurIPS Call for Papers”
> >
> > To be clear: I do not want to dismiss the authors' approach nor the work that went into it. I am not opposed to application papers. But application papers published in the Neuroscience and Cognitive science track do tend to exhibit methodological novelty even when they port over existing approaches, and/or they show how an existing approach can produce novel findings.
> >
> > As the authors state, the generative model and inference approach have been previously published. The authors have used a diagonal plus low rank ansatz for the Wishart, which is a reasonable, incremental advance. They also report that they have implemented a Poisson observation model, a technical advance that was not reported in the original submission.
> >
> > > Even outside of neuroscience, Wishart processes are a relatively exotic model. Prior applications of the method appear mostly confined to quantitative finance.
> >
> > I am not sure what the argument is here. If the method has not been used and provides benefits for analysis, that's great. If it doesn't provide benefits, it doesn't matter that it's exotic, right?
> >
> > > We hope that the reviewer can understand it is challenging to validate a new form of analysis while simultaneously providing a scientific breakthrough. The primary focus of our paper is methodological, but we have done our best to outline scientific future directions (see our "general rebuttal").
> >
> > Absolutely. But as Reviewer mr49 noted, the results are somewhat underanalyzed, and also, per Reviewer ruJu, they rely mostly on visual inspection for their impact.
> >
> > > Thus, to approximate the posterior we perform joint variational inference over the GP and WP parameters (they do not factorize as pointed out by Reviewer mr49). Each parameter is modeled with a Gaussian with a learnable mean and variance, and no correlations are modeled across parameters.
> >
> > Thank you for the clarification.
> >
> > > Our original statement—"number of estimated parameters grows quadratically... while the number of measurements grows only linearly"—is correct.
> >
> > I apologize if I'm being dense, but if I have $K$ observations of an $N$-vector, I have $K$ pairs of numbers with which to estimate each unique element of the covariance matrix, correct? So the uncertainty of each entry still decreases as $1/K$? The number of parameters grows quadratically (with $N$), but so does the number of paired observations (with $K$). That is, the full covariance matrix is a sum of $N$ symmetric rank-1 matrices: $\boldsymbol{\Sigma} = \sum_{i=1}^n \mathbf{u}\mathbf{u}^\top$, so while it has $\mathcal{O}(N^2)$ parameters, it has no more information than $N$ $N$-vector observations, so one only needs $K \sim \mathcal{O}(N)$. Put another way, while the number of parameters increases with dimension, so does the size of each observation.
> >
> > In fact, the Vershynin paper the authors cite accords with this: For a covariance matrix in $N$ dimensions (in the authors' notation), the necessary number of observations is $K(N) \sim \mathcal{O}(N)$ for sub exponential distributions and $\mathcal{O}(N\log N)$ at worst. That is, the intuition that one needs a specific number of observations _per covariance matrix entry_ for a given level of accuracy is incorrect. The Vershynin paper shows that what is needed is a specific number of observations _per dimension_.
> >
> > > Scientific contribution. We agree with Reviewer mr49’s suggestion that “it would help the significance somewhat if this estimation method could help discover (or refine) a scientific conclusion.” Along similar lines, Reviewer ruJu inquired whether it is possible to “demonstrate predictive power (e.g. a regression or classification task).”
> >
> > I appreciate the authors' clarifications on these points.

---

> > > ### Author Response · Authors · 2023-08-11
> > >
> > > We appreciate your responses. We hope that the technical contributions (low-rank structure and now also Poisson noise model) together with the scientific analyses we highlighted in the general rebuttal have raised your interest.
> > >
> > > > ***I am not opposed to application papers.***
> > >
> > > Thank you for this clarification.
> > >
> > > > ***But application papers published in the Neuroscience and Cognitive science track do tend to exhibit methodological novelty even when they port over existing approaches, and/or they show how an existing approach can produce novel findings.***
> > >
> > > We think the reviewer is being fair. In our view, the level of "methodological novelty" in this track covers a pretty wide spectrum: some papers do propose fairly new models, while others have very little methodological advance. (We would rather not cite specific examples here out of courtesy.)
> > >
> > > Our paper does include technical advances (see below). But we think the real litmus test should be: **Will this paper have a measurable impact on the way people approach neural data analysis?** That is, if this paper didn't exist, would people analyze their data sub-optimally or overlook an opportunity to investigate a certain question?
> > >
> > > Currently, noise correlation analysis is only performed on very simple stimulus sets (e.g. across two oriented gratings in Rumyantsev et al. 2020). This is because people think covariance estimation isn't tractable with few trials per condition, and common methods in neuroscience (e.g. Yatsenko et al. 2015 used Ledoit-Wolf and Graphical LASSO) don't pool power across nearby conditions. We think Wishart processes are a great way to overcome this problem, and unlock the possibility of many new experiments/analyses.
> > >
> > > Our general rebuttal outlines two additional use cases (quadratic decoders and estimating Fisher information) as concrete next steps.
> > >
> > > > ***The authors have used a diagonal plus low rank ansatz for the Wishart, which is a reasonable, incremental advance. They also report that they have implemented a Poisson observation model, a technical advance that was not reported in the original submission.***
> > >
> > > Thank you for recognizing these advances. We agree that they are incremental&mdash;perhaps so much so that we failed to sufficiently highlight the low-rank ansatz in our initial contribution. Nonetheless, these tweaks were important to get the model to work in practice, so we think they are important to be put into the scientific record.
> > >
> > > > ***I am not sure what the argument is here. If the method has not been used and provides benefits for analysis, that's great. If it doesn't provide benefits, it doesn't matter that it's exotic, right?***
> > >
> > > Our point was that neuroscientists are unlikely to notice this model, absent our work. Put differently, it would be fair to criticize our paper for being a "straightforward application" if the method itself was well-known and had a standard implementation in e.g. scikit-learn.
> > >
> > > We agree it is more important to show the model actually provides a benefit to the field. (We think it does.)
> > >
> > > > ***the results are somewhat underanalyzed... they rely mostly on visual inspection***
> > >
> > > Our main results are not visual, but quantitative metrics: cross-validated log-likelihoods, recovery of the ground truth covariance in the operator norm on synthetic data, and now decoding performance (Rebuttal Fig C) and Fisher information (Rebuttal Fig D). We are not sure what is meant by "underanalyzed" but we note that the quoted reviews ultimately concluded that the merits of our paper outweigh its limitations. We hope our individualized responses to those reviews addressed those concerns.
> > >
> > > > ***The Vershynin paper the authors cite accords with this: For a covariance matrix in N dimensions (in the authors' notation), the necessary number of observations is K ~ N***
> > >
> > > We still stand by our original statement. But the upshot is that we will revise the text to be more precise (i.e. we will just cite the result from Vershynin), and we appreciate you helping us fine-tune our message.
> > >
> > > Our original statement was essentially: if $K$ stays constant and $N$ grows, your estimation of covariance can degrade. Intuitively, this is because the number of measurements equals $KN$ (i.e. $K$ observed $N$-vectors), while the number of parameters you need to estimate is $\mathcal{O}(N^2)$. That is, the number of measurements grows linearly with $N$ (for fixed $K$) while the number of parameters grows quadratically.
> > >
> > > Assuming sub-exponential variables, Vershynin says that if you take $K = c N$ (for some constant $c$) you will get good performance. In this case the number of measurements you make is $K N = c N^2$ which is the same order of magnitude as the number of parameters you need to estimate, $\mathcal{O}(N^2)$. Thus, one *does* need roughly the same number of measurements per covariance matrix entry.
> > >
> > > Perhaps our original statement was confusing and the reviewer thought we intended to imply that you'd need $K = N^2$ trials?

---

> > > > ### Comment · Reviewer_yodS · 2023-08-11
> > > >
> > > > > Perhaps our original statement was confusing and the reviewer thought we intended to imply that you'd need $K=N^2$ trials?
> > > >
> > > > The message I took away from reading the sentence was that the scaling was somehow poor. But the required number of observations scales linearly with $N$, which is actually very good performance! I now understand that what the authors are envisioning is a situation with _fixed_ $K$ and $c \sim \mathcal{O}(100)$, which does not give them much of any room to increase $N$. I tend to think of this as a practical limitation rather than a scaling difficulty, but at least I understand the claim now.

---

### Official Review · Reviewer_mr49 · 2023-07-06

**Soundness:** 4 excellent
**Presentation:** 4 excellent
**Contribution:** 3 good
**Rating:** 7
**Confidence:** 3

**Summary:**

The estimation of noise covariance in neuroscience is limited by the experimental difficulty of obtaining large numbers of trials for the same neurons, but also the desirability of large numbers of neurons. Here the authors begin by recognizing this need for lower-variance estimators of the noise covariance, and especially those that leverage correct assumptions specific to this domain. This submission centers around leveraging the fact that noise covariance is often estimated not only by repeating a single stimulus many times, but also by showing very similar stimuli parameterized by a continuous number (such as grating orientation), which should evoke similar responses. To leverage this assumption, the authors build a probabilistic model of neural responses in which assumes that the response statistics vary smoothly with a known parameter x. Specifically, transformed neural responses are modeled as a Gaussian distribution whose mean and factorized covariance ($U$) are Gaussian Processes of x (with a squared exponential kernel). The parameters of this probabilistic model, including the covariance as a function of x, are inferred via mean-field variational inference. The estimation method is validated on simulated data and neural recordings in mouse and macaque.

**Strengths:**

This well-written paper tackles a practical problem already faced in many experimental designs in neuroscience, and will likely find wide use if the code is well-documented. The solution is elegant, offers a nice balance of simplicity and power, and benefits from its specific tailoring of known statistical methods to a particular statistical problem in the domain of neuroscience.

**Weaknesses:**

Although I see the benefits of model simplicity, one wonders what this model might miss about the neural response. It is worth noting that the estimated covariance is, strictly speaking, the estimated covariance of the model's residual error of its mean estimate. If this mean is poorly estimated, the noise covariance will be larger than actuality, if not wholly inaccurate. The accuracy of the mean response, in turn, depends on the assumptions built into the model. How much is it accurate to say that neural responses, in general, have a mean firing rate that is a GP of the stimulus parameter with a squared-exponential kernel? The utility of analyzing the resulting $\Sigma(x)$ depends on how much one trusts that this implied encoding model is a good one.

With this said, it is at least more transparent here that analyzing summary statistics like noise correlations always implies, under the hood, a statistical model of neural responses. It is nice here that this is explicit, and I think a step in the right direction.

Finally, it would help the significance somewhat if this estimation method could help discover (or refine) a scientific conclusion in, say, the Allen dataset. What more can we learn from existing data with a lower-variance covariance estimate?

**Questions:**

I have a request for some analyses that would show how well this model captures neural responses.
 - First, for the presented neural data it would be nice to see well this model compares in log likelihood to a probabilistic baseline that is more expressive than a GP mean. The ELBO of a VAE comes to mind, but there are many options in a good-faith effort to build powerful (if uninterpretable) probabilistic models.
 - It would also be nice to see an analysis of synthetic data where some of the assumptions are broken. For example, what happens on data in which the smoothness parameter is not uniform but is itself a function of the inputs? This seems like a near-certain outcome in real neural responses. (For example, in efficient codes of natural images, certain small changes in inputs result in larger changes in neural activity. As per the oblique effect, one might expect that neural responses vary more quickly w/r/t orientation for the more-frequent cardinal orientations as compared to less-frequent diagonal orientations). A discussion of these figures could help ensure this estimator is used where appropriate.

For clarity, could the graphical model of this generative model and variational inference procedure be presented as a figure?

Not much was said about the intricacies of learning and inference. Figures in this direction would likely help adopters troubleshoot the problems usually found in ELBO-trained methods. E.g. what is the tradeoff of accuracy vs speed in the number of MC samples used to estimate of the gradient of the loss?
 - Related: In Fig 3B, is this the max, min, and mean (across seeds) of the median LL across 30 folds? If so, that would narrow these distributions considerably relative to just the distribution of LL across folds on a single run. This would somewhat obscure the variance of the estimator. A thorough analysis of the variance of this estimator would be interesting, help instill confidence in its use, and align with the paper's main narrative (as it is likely smaller than naïve covariance estimation).

Might the kernel be learned via a more expressive function? It strikes me that contrastive learning methods learn such kernels (and are often talked about as learned distance metrics) and could be readily slotted into this framework without (in my opinion) much loss in interpretability. A transfer-learning/ foundation model approach would mitigate the large sample size requirement of a large model like this. There was a sentence about this in the Discussion, so count this as a vote of excitement in that future direction.


**Limitations:**

Some limitations were discussed. Restating some things above, it’d be nice to see more acknowledgements of the limitations relating to the implied GP encoding model and the potential shortcomings of the nonconvex, ELBO-based inference method.

---

> ### Author Rebuttal · Authors · 2023-08-09
>
> > **Although I see the benefits of model simplicity [...].**
>
> We agree that accurately inferring the mean is an important aspect of accurately inferring the covariance. Using a GP to enforce some smoothness in the mean response across conditions is indeed reasonable. See, for example: Wu et al. ("Gaussian process based nonlinear latent structure discovery in multivariate spike train data.", NeurIPS 2017).
>
> The details of the model will likely depend on the particular neural dataset under study. It is certainly possible that other kernels are better suited to certain situations. Our code package will of course have the standard kernels implemented so that users can try different forms of the model.
>
> While we think this is a reasonable first pass, it is easy to swap in new kernel functions, methods for kernel learning, or even methods for non-stationary Gaussian process inference as discussed below. We are excited to try these new directions in future work.
>
> > **[...] it is at least more transparent here [...].**
>
> Thank you. We agree with these sentiments.
>
> > **[...] discover (or refine) a scientific conclusion [...].**
>
> Thank you for this comment, which motivated us to flesh out how WPs can provide estimates of Fisher Information that are (a) more accurate, and (b) come with Bayesian uncertainty estimates. Fisher Information is a fundamental quantity in theoretical neuroscience that is used to characterize thresholds of perception (see Moreno-Bote et al., 2014, Nature Neuroscience). Please see the general rebuttal and Rebuttal Fig D for more information.
>
> > **[...] probabilistic baselines.**
>
> Thank you for the interesting suggestion. In our initial discussions, we were not able to find a straightforward way of performing this experiment. Popular VAEs consider a factorization in the latent space which is independent across different dimensions, estimating a diagonal noise covariance in the latent space as opposed to the high-dimensional firing rate space.
>
> Deep VAEs are data-hungry (require many trials) while the applications we are interested in are very trial-limited. We also feel that doing a comprehensive comparison to VAEs is outside the scope of this paper since VAEs represent a very broad class of models with different objective functions (e.g. beta-VAE), architectures, and hyperparameters. Please let us know if you have a concrete model in mind that could be implemented in the time frame of the review process. We are happy to try.
>
> > **[...] some of the assumptions are broken [...].**
>
> We thank the reviewer for the insightful suggestion. Indeed model misspecification is likely to occur for any model of neural data. We would like to re-emphasize that a GP or WP prior does not necessarily mean that the neighboring conditions are equi-distanced in the mean or covariance space. The prior encourages the nearby conditions to have close means and covariances, once the model is conditioned on the observed data, it allows for the distances to be influenced by the empirical values. In technical terms, GPs with stationary kernels are asymptotically consistent (see the introduction of Koepernik et al. "Consistency of Gaussian process regression in metric spaces." for a review).
>
> Regarding the suggestion, again our initial discussion was not conclusive about how to perform this experiment. GPs with stationary kernels do not allow for a varying smoothness parameter. One way to incorporate the reviewer’s suggestion would be to use non-stationary kernels, but choosing a non-stationary kernel that’s consistent with neuroscientific observations is not straightforward. We believe our work opens up the possibility of adapting non-stationary Gaussian process methods to this problem, and we now discuss this as a future direction in our revision. See Paun et al. "Stochastic variational inference for scalable non-stationary Gaussian process regression." for more information.
>
> > **For clarity, could the graphical model [...]?**
>
> Please see Rebuttal Fig 1G,H. We will update the manuscript with this information.
>
> > **[...] intricacies of learning and inference [...].**
>
> In our code package, all the parameters for every run of the algorithm are stored in configuration (yaml) files allowing for inspecting and modifying the parameters. An example config file is included in the uploaded code package attached to the supplementary.
>
> Fortunately, we have not observed sensitivities to the optimization parameters. We’ve used Adam optimizer with a learning rate of 0.001 with a single MC sample and 10000 iterations for all experiments. In our experiments, using multiple MC samples did not lead to qualitatively different outcomes. As expected, the run time is moderately slower with multiple MC samples; overall run time is still on the order of a few minutes, so this is not a major concern. The number of MC samples is a tunable parameter in our code package.
>
> > **Related: In Fig 3B, is this the max, min [...].**
>
> We are not 100% certain what the reviewer is requesting. We reported the median log-likelihood of individual observations in the test set, which is similar to the mean. We plan to substitute the mean log-likelihood in our revision since it produced essentially identical results (see Rebuttal Fig E,F).
>
> The variance of the LL across individual samples is not related to the variance of the estimator. The variance of an estimator is a frequentist notion of uncertainty that does not neatly map onto the Bayesian approach that we’ve adopted here. We don’t think that the variance of LL across individual samples is usually of interest to report, but perhaps we are missing something or misinterpreting the reviewer’s request. We are happy to continue the discussion.
>
> > **[...] more expressive function [...].**
>
> We thank the reviewer for sharing our excitement. Indeed this is a direction that we are considering and excited about for future work.

---

> > ### Comment · Reviewer_mr49 · 2023-08-16
> > **Thanks for the changes**
> >
> > Hi, I believe my concerns are fully addressed. On reconsideration, I don't believe model misspecification is as scary of a threat as I first considered it. In this application domain (of smoothly parameterized stimuli) a GP assumption is a good start. This also also relates to my request for expressive baselines, which I agree won't be so feasible in these stimulus-limited domains. Such concerns are likely more relevant for (future) applications of these ideas to naturalistic stimuli and learned kernels.
> >
> > In the revision I especially appreciate the new application to Fisher Information. This will certainly be of interest to those at the intersection of psychophysics and neural physiology.
> >
> > In light of these changes I will raise my score by 1 to 7.

---

### Official Review · Reviewer_sHno · 2023-07-06

**Soundness:** 3 good
**Presentation:** 3 good
**Contribution:** 3 good
**Rating:** 7
**Confidence:** 2

**Summary:**

- Exhibits good performance, especially for recordings of large neural populations with very few trials per condition.
- Enables dense sampling of conditions with only a single trial per condition, unlike standard estimators that require a large number of trials per condition (although based on empirical assumptions).

**Strengths:**

- Exhibits good performance, especially for recordings of large neural populations with very few trials per condition.
- Enables dense sampling of conditions with only a single trial per condition, unlike standard estimators that require a large number of trials per condition (albeit grand empirical estimator).

**Weaknesses:**

- Computationally more expensive than standard covariance estimators

**Questions:**

- Figure 2: What does the "Wishart oracle" mentioned in the legend represent? Although it is mentioned, it does not appear to be shown in any panel. Additionally, why is "Empirical" only plotted in panels D3 and D4? It is assumed that its log probability in panel C diverges, but what about D1 and D2? Furthermore, why does the mean operator norm not align with the log probability for C3 and D3? In D4, one would expect the gap between "Grand Empirical," assuming infinite smoothness, and Wishart to consistently widen with lower smoothness.
- Figure 4: How is it possible for condition 1 to be present in both the training conditions (panel C) and the test conditions (panel D)?
- What is the computational runtime compared to standard covariance estimators, particularly in comparison to the "Grand Empirical" estimator that performs well on the real experimental data shown in Figures 3 and 4?

**Limitations:**

The authors have discussed limitations of their work.

---

> ### Author Rebuttal · Authors · 2023-08-09
>
> > **Computationally more expensive than standard covariance estimators**
>
> We have clarified that the computational burden is reasonable: it takes 80 seconds to fit a dataset with 100 neurons, 80 conditions, 32 trials per condition. Notice that we run 10,000 iterations of our optimization algorithm, therefore the run time for such a dataset is about 8 milliseconds per iteration. The runtime for other algorithms for a similarly sized dataset are the following (in seconds): PoSCE: 45, Graphical Lasso: 24, Ledoit-Wolf: 0.3, and Empirical: 0.3.
>
> Experimental data in neuroscience is extremely expensive and time-consuming to collect. Equipment and facilities alone cost tens of thousands of dollars, and animals may need to be trained for weeks (if not months) to perform certain tasks. It is also ethically imperative to use as few animal subjects as possible (particularly for non-human primates). In short, the experimental datasets we target are extremely valuable: neuroscientists are highly motivated to perform high-quality analyses, even if they require marginally more computational resources.
>
> We also wish to remind the reviewer that our model has fundamentally new capabilities that are not present in simple baselines. In particular, we can infer a continuous manifold for the mean and covariance of neural responses (see Fisher Information analysis in Rebuttal Fig 1D), generalize to entirely unseen conditions (see Fig. 3D), and quantify our uncertainty in a Bayesian framework. A fair comparison between inference times should take this difference into account.
>
> > **Figure 2: What does the "Wishart oracle" mentioned in the legend represent? Although it is mentioned, it does not appear to be shown in any panel.**
>
> Thank you for noticing. This corresponds to an experiment that we removed before submission but forgot to remove from the legend. Briefly, this corresponded to a Wishart process but with parameters initialized to the ground truth. Our intention was for this to be a sanity check – an “upper bound” on our model’s performance. However, we intend to remove it in the final version for a streamlined presentation.
>
> > **Additionally, why is "Empirical" only plotted in panels D3 and D4? It is assumed that its log probability in panel C diverges, but what about D1 and D2?**
>
> This has been an oversight, we apologize for this. We have included the revised figures in Rebuttal Fig 1A,B. However, empirical covariance matrices were ill-conditioned and did not produce proper log-likelihood estimates to be included in Fig. 2 C1-4.
>
> > **Furthermore, why does the mean operator norm not align with the log probability for C3 and D3?**
>
> They are measuring performance in two different ways, so it is possible for a model to outperform another in log probability but vice versa on the covariance operator norm. Generally, these two measures of performance are qualitatively aligned (except for the portions of C3 and D3) that the reviewer notes. We will add a short explanation to section 3.1 in the revision.
>
> > **In D4, one would expect the gap between "Grand Empirical," assuming infinite smoothness, and Wishart to consistently widen with lower smoothness.**
>
> We agree. This is generally what we observe in D4. The reviewer may be noting there is an exception at very low degrees of smoothness. Here, all models perform badly since there is no good way to share information across conditions and there are too few trials per condition. It is curious that Grand Empirical “beats” the WP here, which may be due to the variational inference getting caught in sub-optimal local minima. Ultimately, we think this edge case is not a regime of primary interest, since no model performs well (i.e. covariance estimation is not tractable).
>
> > **Figure 4: How is it possible for condition 1 to be present in both the training conditions (panel C) and the test conditions (panel D)?**
>
> There are two separate analyses. In panel C, we train on all conditions and test on a subset of trials per condition. In panel D, we train on a subset of conditions and test on held-out conditions. We show condition 1 in each panel since it was held out in the second analysis. We will update the legend titles and figure legend to make this more clear.
>
> > **What is the computational runtime compared to standard covariance estimators, particularly in comparison to the "Grand Empirical" estimator that performs well on the real experimental data shown in Figures 3 and 4?**
>
> Please see our response above. In short, we believe the computational burden is negligible. Our revision will include a table showing the computational scaling of our algorithm as a function of the number of neurons, conditions, and trials per condition. Roughly speaking, we expect computation times to be on the order of minutes for large-scale neural data.

---

> > ### Comment · Reviewer_sHno · 2023-08-17
> >
> > Thank you for providing a thorough rebuttal. I retain my already positive score.

---

### Author Rebuttal · Authors · 2023-08-09

# General Response
We thank the reviewers for their thoughtful and productive critiques which did not identify any major technical errors. We have done our best to incorporate all reviewer feedback and requests for more details (see individual responses to each reviewer).

Below we summarize the two most important considerations that we hope the reviewers and AC take into account during their final deliberation.

**Technical contribution.** The reviewers largely agreed that our paper addresses an important open problem, is well-written, and is germane to the NeurIPS audience. One potential exception is Reviewer yodS, who writes “this is a very straightforward application of an existing method, which may be novel to neuroscientists, but does not represent much of new conceptual contribution.” We would like to highlight the following:

- The NeurIPS call for papers invites papers on “Applications.” It is especially common for statistical neuroscientists to publish methodological papers in the “Neuroscience and Cognitive Science” conference track. By adapting an existing method to neural data, we believe our paper fits squarely within this tradition.

- Even outside of neuroscience, Wishart processes are a relatively exotic model. Prior applications of the method appear mostly confined to quantitative finance.

- We were not able to identify an implementation of this model that could be used off-the-shelf on our data. Indeed, the Wilson and Ghahramani paper cited by the reviewer used a different inference method (MCMC). We found it necessary to use a more recent variational inference procedure that appeared at NeurIPS in 2019 by Heaukulani & van der Wilk. We found it necessary to implement their model from scratch. A deliverable outcome of our paper will be a self-contained code package that is customized to handle neural data (see supplement for a lightweight version of this package).

- Most importantly, we found that an extension of the model with low-rank-plus-diagonal structure outperformed the full-rank Wishart process. We have not seen such an extension in prior work, even though it has multiple benefits: the learnable diagonal term ensures that matrices are well-conditioned during inference, has fewer parameters (discourages overfitting), and is a reasonable prior for a “spiked covariance” model. If the reviewers agree this is novel, we will revise the text of our paper to emphasize this technical innovation.

- Since our original submission, we have also obtained good results using a Wishart process with Poisson-distributed observations (see Rebuttal Fig E). This is an important consideration for neural data, as evidenced by many prior submissions to the neuroscience track at NeurIPS (e.g. Zhou & Wei 2020 “pi-VAE”). Again, we are unaware of any paper that describes this variation of a Wishart process.

In summary, we respectfully disagree with the notion that our work is merely a “straightforward application.” It involved implementing the model from scratch, adapting the model (e.g. with low-rank-plus-diagonal structure), and benchmarking the model against multiple baselines and across two datasets spanning different species and experimental paradigms. Publishing this effort is necessary to spark an interest in this class of under-appreciated models. Indeed, every colleague in computational neuroscience that we’ve asked is unaware of Wishart process models.

**Scientific contribution.** We agree with Reviewer mr49’s suggestion that “it would help the significance somewhat if this estimation method could help discover (or refine) a scientific conclusion.” Along similar lines, Reviewer ruJu inquired whether it is possible to “demonstrate predictive power (e.g. a regression or classification task).”

We have thus far pursued two scientific applications with promising results.

First, by inferring condition-specific covariance matrices, our method enables decoding conditions by quadratic discriminant analysis (QDA). Linear discriminant analysis (LDA) is a more common model in neuroscience, but quadratic decoders are biologically plausible and of interest to the field (Pagan et al., 2018, Neural Computation). In Rebuttal Fig C, we show that the QDA models enabled by our method outperform linear decoders. This directly answers ruJu’s query.

Second, an even more exciting direction is to use Wishart processes to quantify information-limiting noise correlations (Moreno-Bote et al. 2014, Nature Neurosci). Our method enables continuous interpolation of the mean and covariance across conditions which is necessary to compute the Fisher information. Furthermore, when a Gaussian process is differentiable, its derivative is also a Gaussian process. Thus, we can compute a posterior to quantify uncertainty in our estimate of Fisher information. We showcase our method in Rebuttal Fig D.

Together, the two results above provide a more concrete glimpse into how our methods can be used for scientific discovery. We hope the reviewers are sympathetic to the challenges of providing a complete scientific story in the same paper as an important methodological advance.

**Gameplan.** We will update Figure 2 with the missing lines pointed out by reviewer sHno (see Rebuttal Fig A-B). We will add Rebuttal Figs C and D to Figure 2 of the paper to demonstrate scientific applications and significance. We will describe the Poisson model in a supplemental section (see Rebuttal Fig E). We will swap Rebuttal Fig F in for Figure 3B to address a comment by reviewer ruJu. We will add the graphical model schematic (Rebuttal Fig G) to Figure 1. We will edit the text of the paper as described in the individual reviewer responses. We hope that the reviewers and AC will agree that these edits will substantially improve the manuscript without fundamentally changing the main story and results.

---

### Decision · Program_Chairs · 2023-09-21

**Decision:**

Accept (poster)

**Comment:**

The reviewers were unanimous in their appraisal of this paper's contributions as above the bar for acceptance to NeurIPS, and I congratulate the authors on their detailed and comprehensive rebuttals, which were critical in leading some reviewers to raise their scores.  I'm pleased to report that this paper has been accepted to NIPS.  Congratulations!  Please revise the manuscript to address all reviewer comments and questions.